

# Integrable Matrix Product States from boundary integrability

**Balázs Pozsgay[1,2*], Lorenzo Piroli[3] and Eric Vernier[4]**

**1** Department of Theoretical Physics, Budapest University of Technology and Economics,
1111 Budapest, Budafoki út 8, Hungary
**2** BME Statistical Field Theory Research Group, Institute of Physics,
Budapest University of Technology and Economics, 1111 Budapest, Budafoki út 8, Hungary
**3** Max-Planck-Institut für Quantenoptik, Hans-Kopfermann-Str. 1, 85748 Garching, Germany
**4** The Rudolf Peierls Centre for Theoretical Physics,
Oxford University, Oxford, OX1 3NP, United Kingdom.

* pozsgay.balazs@gmail.com

## Abstract

We consider integrable Matrix Product States (MPS) in integrable spin chains and show that they correspond to "operator valued" solutions of the so-called twisted Boundary Yang-Baxter (or reflection) equation. We argue that the integrability condition is equivalent to a new linear intertwiner relation, which we call the "square root relation", because it involves half of the steps of the reflection equation. It is then shown that the square root relation leads to the full Boundary Yang-Baxter equations. We provide explicit solutions in a number of cases characterized by special symmetries. These correspond to the "symmetric pairs" $(SU(N), SO(N))$ and $(SO(N), SO(D) \otimes SO(N-D))$, where in each pair the first and second elements are the symmetry groups of the spin chain and the integrable state, respectively. These solutions can be considered as explicit representations of the corresponding twisted Yangians, that are new in a number of cases. Examples include certain concrete MPS relevant for the computation of one-point functions in defect AdS/CFT.

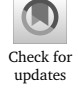

# 1   Introduction

In the past decade considerable effort was devoted to the study of non-equilibrium dynamics of integrable models [1]. One of the main questions was equilibration and thermalization in closed integrable systems [2, 3], and a common setting to study these problems has been the quantum quench. By definition, quenching is a sudden change of certain parameters of the Hamiltonian, and in the simplest case this means that the ground state of some other (integrable or non-integrable) model is released to evolve according to the post-quench integrable Hamiltonian. However, the condition of having a concrete pre-quench Hamiltonian can be relaxed, and we can also study time evolution started from other well-defined states, which are experimentally realizable.

Regarding global quenches one of the central questions was whether the system relaxes to a steady state that can be described by the so-called Generalized Gibbs Ensemble (GGE) [4]. Whereas there is no model-independent answer to this question, the answer was found to be positive in models equivalent to free bosons/fermions [5–10] and in the XXZ spin chain, which is one of the simplest yet most important interacting solvable systems [11]. The status of the GGE for models with higher rank symmetries has not yet been clarified.

The main point of the GGE is that it can describe the steady states arising after quenches from initial states satisfying the cluster decomposition principle. However, its predicting abilities are somewhat limited by its generality. In order to give exact predictions one needs to know the Lagrange-multipliers associated to all conserved charges. These are independent and need to be computed separately (for the precise statements in the XXZ chain see [11–13]).

On the other hand, a different approach was suggested in [14] (see also [15–17]): one should study quenches from those initial states that show certain signs of integrability themselves, thus making the exact computation of the time evolution more tractable. This class of states was called "integrable initial states", and a number of unifying properties were collected in [14]. Perhaps the most important one is that the overlaps between the eigenstates and the initial states can be expressed in a simple factorized form [18–26]. This allows for an exact treatment of the quantum quench, through the Quench Action [27–29] or the Quantum Transfer Matrix [14] methods. In practice the exact solvability means that there is no need to compute the Lagrange multipliers of the GGE (or Bethe root densities of the various particle types) separately.

The factorized overlaps with the integrable states are non-zero only for parity symmetric eigenstates (to be more precise, for Bethe states which are eigenstates of the space reflection operator). On the level of Bethe rapidities this leads to the "pair structure": the set of rapidities consists of pairs with opposite sign. Also, it follows that these initial states are annihilated by all odd (under space reflection) conserved charges of the model. It was argued in [14] that this annihilation property is the most general unifying feature of integrable states, and it should be used as a definition. An important reason for choosing the annihilation property as a definition was that it is relatively easy to check it for a specific state.

In [14] it was also shown that in spin chains there is a direct relation between integrable initial states and the theory of integrable boundaries: every solution to the so-called Boundary Yang-Baxter (BYB) relations can be used to construct a two-site product state which is integrable according to the new definition. This relation can be considered the lattice counterpart of a similar correspondence in integrable QFT, which was worked out in the pioneering work of Ghoshal and Zamolodchikov [30]. The work [14] only treated models with crossing symmetry (mainly the XXZ chain and its higher spin counterparts), where the elements of the so-called $K$-matrices are translated into the real space two-site block using the crossing matrix. In the XXZ chain all two-site states can be obtained this way. Our further papers [31, 32] studied two-site states in the SU(3)-symmetric chain, which lacks crossing symmetry. In such models there are two types of integrable boundary constructions, that are obtained from the conventional ("untwisted") and the twisted Boundary Yang-Baxter relation. In [31, 32] we showed that the local two-site states are always related to solutions of the latter.

An important consequence of the relation with boundary integrability is that the Loschmidt amplitude can be computed using analytic methods [33–36]. On the one hand, this leads to an independent method to find the rapidity distribution functions characterizing the post-quench steady state [31, 32, 34]. On the other hand, it allows one to prove the validity of important properties of the latter, such as the $Y$-system relations for the rapidity distribution functions [34], which were previously only conjectured [37]. Furthermore, this correspondence can be used to determine the thermodynamic part of the overlaps, which in the XXZ chain led to a conjecture for the overlaps with arbitrary two-site states [20].

Integrable initial states have been studied also in QFT's, see for example [15, 38–42]. An early finite volume overlap formula (having the same structure as the later results) already appeared in [43].

In a completely independent line of development, integrable initial states were also found in the context of the AdS/CFT conjecture [21–26]. One-point functions in theories with a defect are given by overlaps with eigenstates of some local spin chain and certain Matrix Product States (MPS) constructed from the generators of group symmetries. It was found that all corresponding overlaps display the pair structure, even in the higher rank models that are solved by the nested Bethe Ansatz. Moreover, the overlaps have a factorized form: they are given by polynomials of $Q$-functions and a ratio of the so-called Gaudin-like determinants. The pair structure implies integrability of these states, which was proven by an independent method in [26].

Despite the increasing amount of information about the new integrable MPS their relation with boundary integrability remained unknown. In [14] it was shown that in the XXZ chain all known integrable MPS are obtained by the action of transfer matrices on two-site states, thus they fit into the framework of the BYB. However, in models with higher rank symmetries this question remained unanswered.

This is the problem that we intend to treat in the present paper. We study the connection between the integrability condition and the twisted BYB relation, and construct new explicit solutions that produce the integrable states found in AdS/CFT. As a byproduct, we also obtain new integrable MPS's.

Our solutions of the twisted BYB involve an additional degree of freedom corresponding to the auxiliary space of the MPS, thus they are "operator-valued solutions". The twisted BYB relation (supplied with a certain symmetry relation) serves as the defining relation of the twisted Yangian [44, 45]. Our solutions are thus explicit representations of the twisted Yangians of various types. The representation theory of these twisted Yangians has been studied in detail in [46–48]. In the main text we explain how our results fit into this framework and show that our explicit realizations are new in many cases.

The structure of the paper is as follows. In Section 2 we introduce the models and some examples for the integrable MPS to be studied. In 3 we investigate the integrability condition and show how it is related to the twisted Boundary Yang-Baxter relation. Here we also introduce a new linear intertwining relation called the "square root relation". In 4 we explain the connection between our formalism and the defining relations of the twisted Yangian, and we also discuss the known results about the representations of the latter object. Sections 5 and 6 include explicit solutions for the intertwining relation; these solutions describe the integrable MPS introduced earlier. We conclude in 7, and two simple computations are described in Appendices B and C.

## 2   Integrable MPS

In this work we deal with integrable lattice models associated to specific Yang-Baxter algebras, possessing certain group symmetries. The central object is the fundamental $R$-matrix acting on the tensor product of two vector spaces $V_1 \otimes V_2$. We will be interested in cases when the $R$-matrix is symmetric with respect to of classical Lie-group $\mathcal{G}$, and the local physical vector spaces carry the defining representation. We will consider the $\mathcal{G} = \mathrm{SU}(N)$ invariant case [49, 50]

$$R(u) = \frac{P + u}{1 + u} \tag{1}$$

and the $\mathrm{SO}(N)$ invariant case [50]

$$R(u) = \frac{(u + c)P + u(u + c) - uK}{(u + c)(u + 1)}. \tag{2}$$

Here $P$ is the permutation operator, and $K$ is the so-called trace operator with matrix elements $K_{ab}^{cd} = \delta_{ab}\delta^{cd}$ and $c = N/2 - 1$. We limit ourselves to these two series of symmetry groups, because their relevance to quantum many body physics [51][1] and to the AdS/CFT correspondence [53].

Both $R$-matrices satisfy the unitarity condition

$$R(u)R(-u) = \mathbb{1} \tag{3}$$

and the Yang-Baxter relation

$$R_{23}(v - z)R_{13}(u - z)R_{12}(u - v) = R_{12}(u - v)R_{13}(u - z)R_{23}(v - z), \tag{4}$$

which is understood as an equation in $\mathrm{End}(V_1 \otimes V_2 \otimes V_3)$ and $R_{jk}$ acts on the spaces $j$ and $k$.

We will also frequently use the matrix $\check{R}(u) = PR(u)$, which satisfies

$$\check{R}_{12}(v - z)\check{R}_{23}(u - z)\check{R}_{12}(u - v) = \check{R}_{23}(u - v)\check{R}_{12}(u - z)\check{R}_{23}(v - z). \tag{5}$$

---

[1] Continuum models with $\mathrm{SU}(N)$ symmetry have been studied in experiments with cold atomic gases [52]. In this paper we only treat lattice models, but they can be used as a starting point towards the continuum models, which can be obtained by certain scaling limits.

Our $R$-matrices also satisfy the permutation symmetry

$$PR(u)P = R(u). \tag{6}$$

We put forward that our methods and results can be extended to models where (6) does not hold, for example to the so-called Perk-Schulz model associated to the quantum group $U_q(\mathfrak{sl}(N))$. Nevertheless here we assume (6), which will imply some minor simplifications in some of the computations.

Group invariance of the $R$-matrices means that for any $G \in \mathcal{G}$

$$(G_1 \otimes G_2)R(u) = R(u)(G_1 \otimes G_2). \tag{7}$$

Here it is understood that the group element $G$ acts in the defining representation in the spaces $V_1$ and $V_2$. Sometimes we will loosely denote by $\mathcal{G}$ the defining representation as well.

A special role will be played by the partial transpose operation. In both cases (1) and (2) we define

$$R^{\mathsf{T}}(u) \equiv R^{\mathsf{T}_1}(u) = R^{\mathsf{T}_2}(u), \tag{8}$$

where $\mathsf{T}_{1,2}$ denote partial transposition with respect to the first or second vector spaces; their equality follows from the explicit forms of the $R$-matrices. The group invariance properties of $R^{\mathsf{T}}(u)$ are obtained from (7) after partial transposition, for example

$$\left(G_1 \otimes \left(G_2^{\mathsf{T}}\right)^{-1}\right)R^{\mathsf{T}}(u) = R^{\mathsf{T}}(u)\left(G_1 \otimes \left(G_2^{\mathsf{T}}\right)^{-1}\right). \tag{9}$$

In the SO($N$) symmetric case we have $\left(G^{\mathsf{T}}\right)^{-1} = G$ for every group element, thus $R^{\mathsf{T}}(u)$ is also group invariant. Moreover, the matrix satisfies the following crossing relation:

$$R^{\mathsf{T}}(u) = \frac{(u+c-1)u}{(u+c)(u+1)}R(-u-c), \qquad c = N/2 - 1. \tag{10}$$

This also implies that $u = -c/2$ is a crossing symmetric point, where

$$R^{\mathsf{T}}(-c/2) = R(-c/2). \tag{11}$$

On the other hand, in the SU($N$)-symmetric case we get the group symmetry property

$$(G_1 \otimes G_2^*)R^{\mathsf{T}}(u) = R^{\mathsf{T}}(u)(G_1 \otimes G_2^*), \tag{12}$$

where the asterix denotes complex conjugation. We can thus interpret that $R^{\mathsf{T}}(u)$ acts on the tensor product of a defining and a conjugate representation of SU($N$).

A further important property of the $R$-matrices is the initial condition

$$\check{R}(0) = \mathbb{1}, \tag{13}$$

which is satisfied in both cases.

We consider spin chains of length $L$ with Hilbert spaces $\mathcal{H} = \bigotimes_{j=1}^{L} \mathbb{C}^N$. We define the monodromy matrix of a homogeneous spin chain of length $L$ as an operator acting on $\mathcal{H} \otimes V_0$,

$$T(u) = R_{0L}(u)\dots R_{02}(u)R_{01}(u). \tag{14}$$

Here 0 refers to the so-called auxiliary space and $V_0 \approx \mathbb{C}^N$. The transfer matrix is the trace over the auxiliary space:

$$t(u) = \mathrm{Tr}_0 T(u). \tag{15}$$

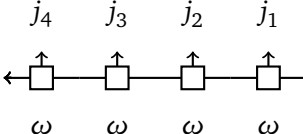

Figure 1: A pictorial representation of the Matrix Product State (21) built from the one-site block $\omega$. Here the outgoing indices $j_k = 1,\ldots,N$, $k = 1,\ldots,L$ represent the physical degrees of freedom, and the horizontal lines denote the action of the matrices $\omega_{j_k}$. The matrices are assumed to act from the right to the left; the arrow on the leftmost horizontal link signals this convention. The trace in the definition (21) implies periodic boundary conditions.

Due to (5) the transfer matrices form a commuting family of operators, and they can be used to define the local conserved charges of the model as

$$Q_j = \left(\frac{d}{du}\right)^{j-1} \log t(u)\bigg|_{u=0}, \qquad j = 2, 3, \ldots . \tag{16}$$

With these conventions $Q_j$ is a sum of local operators spanning $j$ sites at most, and specifically $Q_2$ can be identified with a local two-site Hamiltonian. With these normalizations we get in the SU($N$) case

$$H = Q_2 = \sum_{j=1}^{L} (P_{j,j+1} - 1), \tag{17}$$

whereas in the SO($N$) case

$$H = Q_2 = \sum_{j=1}^{L} (P_{j,j+1} - K_{j,j+1}/c - 1), \tag{18}$$

and periodic boundary conditions are understood in both cases.

The behaviour of the charges under space reflection is

$$\Pi Q_j \Pi = (-1)^j Q_j, \tag{19}$$

where $\Pi$ is the space reflection operator. For further use we also introduce the space reflected transfer matrix:

$$\tilde{t}(u) = \Pi t(u) \Pi = \text{Tr}_0 \, R_{01}(u) \ldots R_{0L}(u). \tag{20}$$

In this work we discuss translationally invariant Matrix Product States (MPS's). Let us take $N$ $d$-dimensional matrices $\omega_j$, $j = 1 \ldots N$, acting on a further, $d$ dimensional auxiliary space $V_A$ (not to be confused with the auxiliary space used in the definition of the transfer matrices). The MPS is an element of the Hilbert space of the chain defined as

$$|\Psi_\omega\rangle = \sum_{j_1,\ldots,j_L=1}^{N} \text{Tr}_A \big[ \omega_{j_L} \ldots \omega_{j_2} \omega_{j_1} \big] |j_L,\ldots,j_2,j_1\rangle. \tag{21}$$

Here $|j_L,\ldots,j_2,j_1\rangle$ are the real space basis vectors with $j_k = 1,\ldots,N$, $k = 1 \ldots L$. A graphical interpretation of the MPS is given in Fig. 1. It is important that the vectors (21) are translationally invariant, and the same set of matrices $\{\omega_j\}$ are used for every finite volume $L$. In the rest of the paper we will use the term MPS for both the set $\{\omega_j\}$ and the vectors (21), and sometimes we will use the short-hand $\omega$ to refer to the collection of matrices $\{\omega_j\}$.

An MPS is invariant under a subgroup $\mathcal{G}' \subset \mathcal{G}$ if for every $G \in \mathcal{G}'$

$$\left[ \bigotimes_{j=1}^{L} G_j \right] |\Psi_\omega\rangle = |\Psi_\omega\rangle . \tag{22}$$

This can be achieved by constructing a group invariant $\omega$. Let us assume that there is a representation $\Lambda_\omega$ of $\mathcal{G}'$ acting on the auxiliary space $V_A$. The building block $\omega$ is group invariant, if for all $G \in \mathcal{G}'$

$$\Lambda_\omega(G^{-1})\omega_j \Lambda_\omega(G) = \sum_k G_{jk}\omega_k, \qquad j = 1, \ldots, N, \tag{23}$$

where $G_{jk}$ are the matrix elements of $G$ in the defining representation. This relation immediately implies (22) [2]. In the following we will use the notation $(\mathcal{G}, \mathcal{G}')$ to denote classes of the MPS, where it is understood that $\mathcal{G}$ and $\mathcal{G}'$ describe the group symmetries of the spin chain and the MPS, respectively.

The condition (23) puts a restriction on the representation $\Lambda_\omega$: it means that the scalar representation of $\mathcal{G}'$ has to be present in the decomposition of the triple tensor product $\Lambda_\omega \otimes \bar{\Lambda}_\omega \otimes \Lambda_{\mathcal{G}'}$, where $\bar{\Lambda}$ is the conjugate representation to $\Lambda$ and $\Lambda_{\mathcal{G}'}$ is the representation of $\mathcal{G}'$ obtained from the defining representation of $\mathcal{G}$ after the restriction $\mathcal{G}' \subset \mathcal{G}$. Examples for such group invariant MPS are listed below.

In this paper we will be interested in cases where $(\mathcal{G}, \mathcal{G}')$ is a so-called symmetric pair[3]. The reason for concentrating on these pairs is twofold. First, it is known that all solutions to the (twisted or untwisted) Boundary Yang-Baxter relations are characterized by symmetric pairs, or slight generalizations thereof (for the quantum deformed case, see [54,55]). Second, the known integrable initial states are characterized by symmetric pairs. In this paper we do not attempt a rigorous classification of all possible integrable MPS, instead we consider a few examples for the symmetric pair $(\mathcal{G}, \mathcal{G}')$.

In this work we focus on the special class of integrable MPS. The properties of integrable states were reviewed in detail in [14]. It is a unifying characteristic that they have non-zero overlaps only with the parity-invariant Bethe states of the system, which leads to the so-called pair structure for the Bethe rapidities. This leads to the condition of annihilation by the odd charges of the model:

$$Q_{2j+1}|\Psi\rangle = 0, \qquad j = 1, \ldots . \tag{24}$$

It follows from (19) and (16) that (supplied with two-site shift invariance) this is completely equivalent to

$$t(u)|\Psi\rangle = \tilde{t}(u)|\Psi\rangle . \tag{25}$$

This condition is stronger than simply parity invariance: the relation $\Pi|\Psi\rangle = \pm|\Psi\rangle$ does not imply (25). We stress that (24) and (25) are exact equalities that hold in every finite volume.

In [14] it was suggested that the relations (24)-(25) should serve as definitions of integrable states. Also, it was shown there that in the spin-1/2 XXZ chains the known integrable MPS can be obtained from solutions of the Boundary Yang-Baxter (BYB) relation. The connection between integrability of the state and integrable boundaries is the following.

When having integrable boundaries one usually means boundaries in space, and the goal is to set up a set of commuting transfer matrices, where the so-called boundary $K$-matrices play

---

[2]It can be shown, that given some additional assumptions the converse is also true: If the MPS is such that the group invariance (22) holds in any volume $L$, and the MPS is completely reducible (a property introduced below), then Theorem 1 implies the existence of a similarity transformation which will form a representation of $G'$ on $V_A$, thus proving the local group invariance property (23).

[3]A pair of Lie groups $(\mathcal{G}, \mathcal{G}')$ is called a symmetric pair if there exists an involutive automorphism $\theta$ of $\mathcal{G}$ such that $\mathcal{G}'$ is the subgroup consisting of the $\theta$-invariant elements. An alternative definition on the level of Lie-algebras is that $(\mathfrak{g}, \mathfrak{h})$ is a symmetric pair if $\mathfrak{g}$ can be split as a vector space into $\mathfrak{h} \oplus \mathfrak{f}$ such that $[\mathfrak{h}, \mathfrak{h}] \in \mathfrak{h}$, $[\mathfrak{h}, \mathfrak{f}] \in \mathfrak{f}$ and $[\mathfrak{f}, \mathfrak{f}] \in \mathfrak{h}$. In this splitting $\mathfrak{h}$ and $\mathfrak{f}$ are the eigenspaces of the automorphism $\theta$ with eigenvalues 1 and -1, respectively.

an essential role. They encode the boundary conditions, and they need to satisfy the so-called reflection equations [56], guaranteeing the commutativity of the transfer matrices. In contrast, in our case the integrable initial states can be considered as boundaries in time. They can be interpreted as integrable boundaries if we build classical 2D partition functions (corresponding to a lattice path integral of some 1D quantum problem) and afterwards exchange the role of space and time. This correspondence is a lattice counterpart of the picture in integrable QFT [30]. It is the purpose of the present work (and in particular Section 3) to apply these ideas of [14] to the MPS in the higher rank $SU(N)$ and $SO(N)$ invariant models.

Examples of integrable MPS were found in the context of the AdS/CFT conjecture [21–24, 26], and factorized overlap formulas were also derived there. The integrability of the MPS was proven in [26] by a method similar to our work; nevertheless the relation with the boundary integrability framework remained unexplored. This is the goal that we set in our paper. We specifically focus on the following list of MPS, all of which are integrable according to the definitions (24)-(25).

1. For the symmetric pair $(SU(3), SO(3))$ the MPS is given by the matrices

$$\omega_j = S_j, \quad j = 1, 2, 3, \tag{26}$$

where $S_j$ are the generators of the $SU(2)$-algebra,

$$[S_j, S_k] = i\varepsilon_{jkl}S_l, \tag{27}$$

in a finite dimensional irreducible representation. In the simplest case of spin-1/2 representation we can choose simply $\omega_j = \sigma_j/2$ where $\sigma_j$ are the standard Pauli matrices.

The $SU(2)$-generators are in the adjoint representation, therefore they satisfy (23) with respect to the $SO(3)$ group.

This MPS appeared for the first time in [23], and the quantum quench started from this state (in the spin-1/2 case) was investigated in [57].

2. For the symmetric pair $(SU(N), SO(N))$ with any $N \geq 3$ the MPS is given by the matrices

$$\omega_j = \gamma_j, \quad j = 1, \ldots, N, \tag{28}$$

where $\gamma_j$ are the $N$-dimensional gamma matrices satisfying the (euclidean) Clifford algebra relations

$$\{\gamma_j, \gamma_k\} = 2\delta_{jk}. \tag{29}$$

The commutators of the gamma matrices can be used as generators of the $SO(N)$ Lie-algebra as

$$t_{jk} = \frac{1}{4i}[\gamma_j, \gamma_k]. \tag{30}$$

The generators constitute the adjoint representation of $SO(N)$, and the set of Dirac matrices satisfies (23) with respect to the spinor representation of $SO(N)$.

For $N = 3$ this state coincides with the spin-1/2 member of the previous family. For $N > 3$ it is new.

3. For the symmetric pair $(SU(N), SO(N))$ with any $N \geq 3$ the MPS is given by symmetrically fused Gamma matrices

$$\omega_j = \Gamma_j^{(n)}/2, \tag{31}$$

where $\Gamma^{(n)}$ are defined in the main text in (115). For $N = 3$ they are identical to the first family listed above, for higher $N$ they are new.

4. Non-trivial MPS with a different type of symmetry breaking are found as follows. Consider the symmetric pair $(SO(N), SO(D) \oplus SO(N-D))$ with some $N \geq 3$ and $1 \leq D \leq N$ and the MPS

$$
\omega_j = \begin{cases} \gamma_j & \text{for } 1 \leq j \leq D \\ 0 & \text{for } D < j \leq N, \end{cases} \tag{32}
$$

where now $\gamma_j$ are the $D$-dimensional gamma matrices.

Two examples of this (for $N = 6, D = 3$ and $N = 6, D = 5$) were studied in [24–26], but the generalization to arbitrary $N, D$ is new.

This list does not exhaust all the integrable MPS that were found in the AdS/CFT literature. Symmetrically fused solutions associated with the pairs $(SO(6), SO(3) \otimes SO(3))$ and $(SO(6), SO(5))$ were also studied there, but in the present paper we limit ourselves to the cases given above.

# 3 Intertwining relations

In this section we analyze the integrability condition (25) and its relation to certain intertwining relations and the twisted Boundary Yang-Baxter (BYB) relation. First we need to introduce some mathematical properties of the MPS.

We call an MPS given by $\omega$ irreducible, if there is no proper subspace $V_A' \subset V_A$ which is an invariant sub-space for all $\omega_a$, $a = 1, \ldots, N$. All examples listed at the end of the previous Section are irreducible.

We will make use of the following simple statement, which is an analogue of Schur's lemma:

**Lemma 1.** *If an MPS is irreducible, then any matrix $U \in End(V_A)$ which commutes with all $\omega_a$ is proportional to the identity matrix acting on $V_A$.*

*Proof.* Any matrix $U$ has at least one eigenvector with some eigenvalue $\lambda$. The eigenspace associated to $\lambda$ is an invariant subspace for all $\omega_a$ due to the commutativity, therefore it has to be identical to the full $V_A$. $\square$

Taking two irreducible MPS $\omega_A$ and $\omega_B$ we can construct a new one simply by addition of the auxiliary vector spaces:

$$
\omega_{A+B,a}(u) \equiv \begin{pmatrix} \omega_{A,a}(u) & 0 \\ 0 & \omega_{B,a}(u) \end{pmatrix}, \qquad a = 1 \ldots N. \tag{33}
$$

For the physical vectors (21) we get

$$
|\Psi_{A+B}\rangle = |\Psi_A\rangle + |\Psi_B\rangle, \tag{34}
$$

due to the additivity of the trace in.

We say that an MPS is completely reducible, if it can be written as a direct sum of irreducible pieces. Alternatively, it means that if there is a common invariant subspace $V_x \in V_A$ for all $\omega_j$, then there is a complementary space $V_y$ with $V_A = V_x \oplus V_y$, which is invariant too. This property means that if we have a triangular block diagonal form as

$$
\omega_j = \begin{pmatrix} E_j & F_j \\ 0 & G_j \end{pmatrix}, \tag{35}
$$

then necessarily $F_j \equiv 0$. Note that if we are interested in the physical vectors, then the $F_j$ are irrelevant, because they do not contribute to the trace. However, below we will be interested in linear relations involving the full matrices without dropping any off-diagonal blocks, therefore the complete reducibility will be crucial[4].

Let $\mathcal{A}$ be the matrix algebra generated by the set of matrices $\{\omega_j\}$. The notions of irreducibility and complete reducibility naturally carry over to $\mathcal{A}$. There is a well known statement which usually goes under the name of Burnside's Theorem on matrix algebras: the algebra $\mathcal{A}$ is irreducible, if and only if $\mathcal{A} = \text{End}(V_A)$.

For any MPS we can define the associated transfer matrix[5] acting on $V_A \otimes V_A$ as

$$\mathcal{T} = \sum_{a=1}^{N} \omega_a \otimes \omega_a^*. \tag{36}$$

It is known from the theory of matrix product states that the eigenvalues and eigenvectors of $\mathcal{T}$ describe the periodicity properties, the half-chain reduced density matrix eigenvalues, and the correlation lengths of the MPS [58].

It is shown in [58], that if the MPS is irreducible, then there is a non-degenerate leading eigenvalue $\lambda_{\max}$ of $\mathcal{T}$ such that $\lambda_{\max} \in \mathbb{R}^+$, and $|\lambda_j| \le \lambda_{\max}$ for all other $j$. We say that an irreducible MPS is pure, if $\lambda_{\max}$ is non-degenerate also in magnitude: $|\lambda_j| < \lambda_{\max}$ for all other $j$ (in [58] this was called the C2 property). If the MPS is not pure, then there are some eigenvalues of the form $\lambda_j = \lambda_{\max} e^{2i\pi p/q}$, and the vector (21) can be written as a sum of $q$-site invariant MPS with lower bond dimension. The purity condition thus says that the MPS can not be decomposed into simpler blocks even if we lift the requirement of one-site invariance.

In the theory of Matrix Product States it is a general important question, how unique the actual matrix representations are. Regarding this question we have the following theorem [59,60][6]:

**Theorem 1.** *If two sets of matrices $\{A_j\}$ and $\{B_j\}$ are completely reducible, and they produce the same MPS for all L:*

$$|\Psi_A(L)\rangle = |\Psi_B(L)\rangle, \tag{37}$$

*then there is a simultaneous similarity transformation S connecting the two sets as*

$$A_j = S^{-1} B_j S, \quad j = 1, \dots, N. \tag{38}$$

The proof of the theorem is given in [60] (Theorem 1) using the representation theory of semi-groups. The theorem also follows from Corollary 2.7 of [59]. This latter paper also shows that it is enough to require the equality of MPS at some large enough length $L^*$, the precise value of which is not important for our purposes. A slightly different formulation of the same statement can be extracted also from [58], but the theorem in this form is not stated there.

It is easy to see that the complete reducibility is indeed needed. Let us consider for example the simple case with $N = 1$ and the two matrices being

$$A_1 = \begin{pmatrix} 1 & 0 \\ 0 & 1 \end{pmatrix} \qquad B_1 = \begin{pmatrix} 1 & \alpha \\ 0 & 1 \end{pmatrix}, \tag{39}$$

---

[4]There can be solutions to the Boundary Yang-Baxter relations that are not completely reducible and involve a nonzero $F_j$ block. The treatment of these cases would need more detailed investigations.

[5]This transfer matrix can be defined for any MPS, regardless of integrability. It is not to be confused with the various transfer matrices built from integrable Lax operators and integrable boundary conditions. Nevertheless the so-called Quantum Transfer Matrix defined later in (77) simplifies to $\mathcal{T}^2$ under a special choice of parameters.

[6]We are thankful to Ben Grossmann for his help in finding the relevant literature.

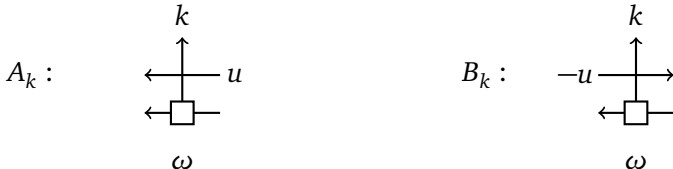

Figure 2: Graphical representation of the dressed matrices defined in (41), which act on the product space $V_0 \otimes V_A$. The outgoing index $k$ stands for the physical degree of freedom. The dressing of the $B_k$ matrices includes a partial transpose, this is denoted by the reversed arrow on the horizontal line. The spectral parameters associated to the horizontal lines are $u$ and $-u$, whereas the vertical line carries 0 rapidity. Here and in the following the $R$-matrices at the crossings are such that their argument is the incoming rapidity coming from the right minus the rapidity from the left. Therefore, both crossings above are described by $R(u)$, but with a different orientation.

with some constant $\alpha \neq 0$. The resulting trace conditions $\mathrm{Tr}A_1^L = \mathrm{Tr}B_1^L$ are satisfied for all $L$, but there is no similarity transformation connecting $A$ and $B$. The reason for this is the invariant subspace and the triangular structure, and that the algebra generated by $B_1$ is not completely reducible.

Now we consider the integrability condition (25). Both sides of the relation (25) can be described by a "dressed" MPS, where the dressing is caused by the action of the two transfer matrices. This is made explicit by defining the corresponding MPS matrices $A_j$ and $B_j$ acting on $V_0 \otimes V_A$, where $V_0$ is the auxiliary space of the transfer matrix and $V_A$ is the space for the action of $\omega_j$. Here $A_j$ corresponds to the original dressing and $B_j$ to the reflected dressing of the MPS. For a physical space $V_1$ let us decompose the $R$-matrix as

$$R_{10}(u) = \sum_{ab} E_{ab} \otimes \mathcal{L}_{ab}(u), \tag{40}$$

where $E_{ab}$ are the basis matrices acting on $V_1$ and the matrices $\mathcal{L}_{ab}(u)$ act on the auxiliary space. Then we have

$$\begin{aligned}
A_j &= \sum_k \mathcal{L}_{jk}(u) \otimes \omega_k \\
B_j &= \sum_k \mathcal{L}_{jk}^{\mathsf{T}}(u) \otimes \omega_k,
\end{aligned} \tag{41}$$

where $T$ denotes simple transpose for each $j, k$.

The $\omega$ matrices satisfy the group invariance (23) under $\mathcal{G}' \in \mathcal{G}$. It follows that the matrices $A_j$ act on the representation $\Lambda_0 \otimes \Lambda_\omega$ of $\mathcal{G}'$, where $\Lambda_0$ is the restriction of the defining representation of $\mathcal{G}$ to $\mathcal{G}'$. On the other hand, the $B_j$ act on $\bar{\Lambda}_0 \otimes \Lambda_\omega$ due to the partial transpose and the group property (12).

The condition (25) says that $\{A_j\}$ and $\{B_j\}$ generate the same MPS for all $L$:

$$|\Psi_A(L)\rangle = |\Psi_B(L)\rangle. \tag{42}$$

This implies:

**Theorem 2.** *If the dressed MPS $\{A_j\}$ and $\{B_j\}$ are completely reducible, then there exists an invertible matrix $K(u)$ which is a simultaneous intertwiner between $\{A_j\}$ and $\{B_j\}$:*

$$A_j K(u) = K(u) B_j, \quad j = 1, \dots, N. \tag{43}$$

*The matrix $K(u)$ acts on $V_0 \otimes V_A$, and it is a function of the rapidity parameter used in the dressing* (41).

*Proof.* This follows immediately from Theorem 1. □

The intertwiner above can be identified with the object called "$K$-matrix" from the theory of boundary integrability. Below we will show that (given some conditions) it satisfies the twisted Boundary Yang-Baxter relation. $K(u)$ intertwines the representations $\bar{\Lambda}_0 \otimes \Lambda_\omega$ and $\Lambda_0 \otimes \Lambda_\omega$ of $\mathcal{G}'$. The representation changing property shows that this object always corresponds to "soliton non-preserving" boundary conditions [61, 62].

It is important to analyze the reducibility conditions for Theorem 2 and their implications. First of all, in those cases when the dressed MPS are irreducible, the $K$-matrices are unique up to an overall phase. If there are invariant subspaces, but the MPS is completely reducible, then the normalization factors of the individual blocks can be chosen independently, and the $K$-matrices are thus not unique.

The above theorem requires complete reducibility, but this does not automatically hold for the dressed MPS (41). A counter-example in the SU($N$)-invariant case is simply the reference state, which can be described by a one-dimensional (scalar) MPS given by

$$\omega_1 = 1, \qquad \omega_k = 0, \text{ for } k \geq 2. \tag{44}$$

Let us denote by $e_k$, $k = 1 \ldots N$ the standard basis vectors. We can see that span$\{e_1\}$ and span$\{e_2, e_3, \ldots\}$ are invariant subspaces for $\{A_j\}$ and $\{B_j\}$, respectively, but the complements are not. In other words, both $\{A_j\}$ and $\{B_j\}$ have a non-trivial triangular structure. However, even in this case there is a non-zero $K$-matrix satisfying (43), which is given explicitly as $K_{11}(u) = 1$, $K_{jk}(u) = 0$ for $j > 1$ or $k > 1$. This $K$-matrix is not invertible, and this reflects the lack of the complete reducibility.

For these cases we have following theorem:

**Theorem 3.** *If the original MPS is built from self-adjoint matrices $\omega_j = \omega_j^\dagger$, then there is at least one solution to the intertwining relation* (43) *with a non-zero $K(u)$.*

*Proof.* Let us take a real $u$ parameter. The definition (41) together with the self-adjointness property implies

$$A_j = B_j^\dagger, \qquad j = 1, \ldots, N. \tag{45}$$

If there is at least one irreducible subspace $V_x$ for the set $\{A_j\}$, then $\{A_j\}$ can always be brought into an upper triangular block form, where the first block corresponds to $V_x$. The self adjointness implies that in this basis the set $\{B_j\}$ will have lower triangular block form. The diagonal block corresponding to $V_x$ is already irreducible, and the complementary block is either irreducible or can be split up into irreducible blocks after further basis transformations. Only the diagonal blocks contribute to the trace, therefore Theorem (1) implies a similarity transformation for each diagonal block separately. However, this does not mean that the full sets $\{A_j\}$ and $\{B_j\}$ are similar. Nevertheless, we can find an intertwiner as

$$K(u) = P_x K_x(u) P_x, \tag{46}$$

where $P_x$ is the projector onto $V_x$ and $K_x(u)$ is the invertible similarity transformation within $V_x$. If $V_x$ is a proper subspace then $K(u)$ is not invertible, nevertheless it satisfies the linear relation (43).

The intertwining relation is analytic in $u$, therefore the solution $K(u)$ can be extended into the complex plane. □

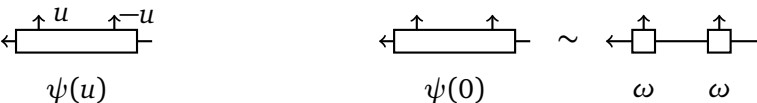

Figure 3: Pictorial representation of the two-site block $\psi(u)$ and its initial condition (see Theorem 4).

This theorem obviously covers the case of the reference state, which was a counter-example to complete reducibility.

The theorem can be extended to those cases when span$\{\omega_j\}$ is self-adjoint, but the matrices themselves are not. This includes all cases listed at the end of the previous Section. We thus conclude that there is a non-zero intertwiner in all of these cases.

The uniqueness of the intertwiner is an important question. We numerically investigated the dressed matrices obtained from some examples of the MPS listed at the end of the previous Section. It was found that in all cases the dressed MPS are irreducible for generic values of $u$. The details of this numerical procedure are explained in Appendix B. We have thus established that the intertwiner is unique for these MPS, but it would be desirable to obtain an analytic proof too.

Let us write the $K$-matrix introduced above as

$$K(u) = \sum_{a,b} E_{ab} \otimes \psi_{ab}(u), \tag{47}$$

where $E_{ab}$ are the elementary matrices acting on $V_0$ and $\psi_{ab}(u)$, $a, b = 1 \ldots N$ are matrices acting on $V_A$. The collection of matrices $\{\psi_{ab}(u)\}_{a,b=1\ldots N}$ will also be denoted simply as $\psi(u)$ and it will be called the two-site block. It can be considered as an element of $\mathbb{C}^N \otimes \mathbb{C}^N \otimes \text{End}(V_A)$. Later in this Section we will use $\psi(u)$ to build inhomogeneous two-site invariant MPS. A pictorial representation is given on the left hand side of Figure 3.

Working out the indices the condition (43) can be written as

$$\check{R}^{de}_{ab}(u)\omega_d\psi_{ec}(u) = \check{R}^{de}_{bc}(u)\psi_{ad}(u)\omega_e, \tag{48}$$

where $a, b, c, d, e = 1 \ldots N$ are the physical indices and summation over $d, e$ is understood. A detailed derivation of (48) is presented in Appendix A. This relation can also be written in a short-hand notation as

$$\check{R}_{23}(u)(\omega \cdot \psi(u)) = \check{R}_{12}(u)(\psi(u) \cdot \omega), \tag{49}$$

which is a relation in $V_3 \otimes V_2 \otimes V_1 \otimes \text{End}(V_A)$. Here and in the following we always understand that the $\check{R}$ matrices act on some of the physical indices (and the numeric indices label the vector spaces), and the actual elements of the equations are matrices acting on $V_A$. A pictorial representation is given in Fig. 4, whereas the interpretation of the intertwiner relation (43) is shown in Fig. 5. We call (49) the "square root relation" (abbreviated as sq.r.r.), because it involves half the steps of the so-called Boundary Yang-Baxter relation, to be introduced below. We stress that (49) is only a short notation for the full relation (48) containing all indices.

**Theorem 4.** *For invertible $\omega_j$ the sq.r.r. implies the initial condition*

$$\psi_{jk}(0) = \omega_j\omega_k, \tag{50}$$

*up to a scalar factor.*

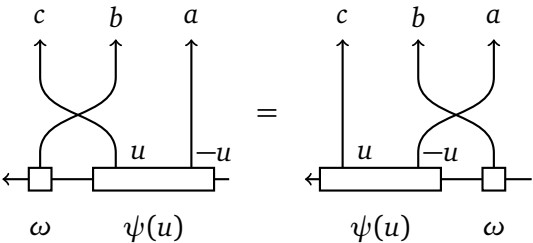

Figure 4: A pictorial interpretation of the "square root relation" (49), which describes the exchange of the two-site block $\psi(u)$ and the one-site block $\omega$. This is a relation in $V_3 \otimes V_2 \otimes V_1 \otimes \mathrm{End}(V_A)$ and the outgoing indices $c, b, a$ describe the basis states in $V_3 \otimes V_2 \otimes V_1$. Fixing $c, b, a$ we obtain matrices acting on $V_A$. The local $\check{R}$ matrices acting at the crossings are defined such that their argument is always the rapidity coming from the right minus the rapidity coming from the left. Thus we get an action of $\check{R}(u)$ on both sides, but on different vector spaces.

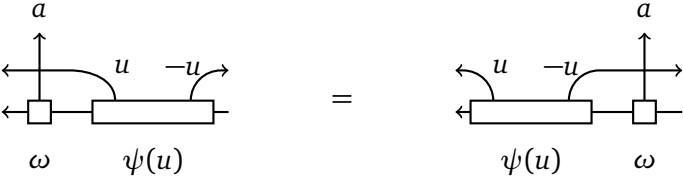

Figure 5: A pictorial interpretation of the "square root relation" as an intertwining relation, see (43). Fixing the outgoing index $a$ we get a relation in $\mathrm{End}(V_0 \otimes V_A)$, where $V_0$ and $V_A$ are the auxiliary spaces of the Lax operators, and the $\omega_j$, respectively. The intertwining relation holds for all $a = 1, \ldots, N$.

*Proof.* From (48) at $u = 0$ we get

$$\omega_i \psi_{jk} = \psi_{ij} \omega_k \qquad \forall i, j, k.$$

So we can write

$$\psi_{j,k} = ((\omega_i)^{-1} \psi_{i,j}) \omega_k = \tilde{\omega}_j \omega_k,$$

where we have defined $\tilde{\omega}_j = (\omega_i)^{-1} \psi_{i,j}$, which has to be independent of $i$.

Substituting back we see that

$$\tilde{\omega}_j \omega_j^{-1} = (\omega_i)^{-1} \tilde{\omega}_i,$$

which is therefore independent both of $i, j$. Introducing $U = (\omega_i)^{-1} \tilde{\omega}_i$ we have

$$\tilde{\omega}_j = U \omega_j = \omega_j U, \tag{51}$$

Using Lemma (1) $U$ is proportional to identity. Up to a re-scaling of $\psi(u)$ we conclude $\tilde{\omega}_j = \omega_j$, thus completing the proof. $\qquad \square$

We also remark that from an alternative point of view, the sq.r.r. can be used to define integrable MPS:

**Theorem 5.** *If there is a solution to* (49) *for some* $\omega$, *such that* $K(u)$ *is invertible for almost all* $u$, *then the MPS built from* $\omega$ *is integrable.*

*Proof.* This follows immediately from the construction: the $K$-matrix intertwines an arbitrary product of the dressed matrices $A_j$ and $B_j$, and after multiplying with $K^{-1}(u)$ and taking the trace this leads to the integrability condition (42). $\qquad \square$

This proof of the integrability of the states is the one-site counterpart of our earlier proof presented in [14], which only treated two-site invariant cases constructed directly from the $K$-matrices. This new proof can be used also in spin chains with odd lengths, where the earlier method was not applicable. The integrability of one-site states at odd lengths was first observed in [20].

If the intertwiner (43) is unique, then it is group invariant:

$$(\Lambda_0(G) \otimes \Lambda_\omega(G))K(u) = K(u)(\Lambda_0(G^*) \otimes \Lambda_\omega(G)). \tag{52}$$

This is easily seen by contradiction: Assuming a non-invariant $K$-matrix it can be seen that after the group transformation it also solves the same relation, and by unicity it has to be proportional to the original $K(u)$. The proportionality factor has to be a one-dimensional representation of the group $G'$, and in our cases all such representations are trivial.

For the two site block $\psi(u)$ the group invariance property takes the form

$$\Lambda_\omega(G^{-1})\psi_{ab}(u)\Lambda_\omega(G) = G_{ac}G_{bd}\psi_{cd}(u). \tag{53}$$

It follows from the Yang-Baxter relations that $\tilde{\psi}(u) = \check{R}(2u)\psi(-u)$ is also a solution to (61), and (13) implies that it satisfies the same initial condition $\tilde{\psi}_{ab}(0) = \psi_{ab}(0) = \omega_a\omega_b$. If the solution of the sq.r.r. is unique, then we obtain the condition

$$\psi(u) = f(u)\check{R}(2u)\psi(-u), \tag{54}$$

with some function $f(u)$ satisfying $f(u)f(-u) = 1$.

Assuming uniqueness, we can always re-define the normalization of $\psi(u)$ so that it satisfies

$$\psi(u) = \check{R}(2u)\psi(-u). \tag{55}$$

This will be called the "symmetry relation". Note that this still leaves room for an arbitrary re-definition $\tilde{\psi}(u) = g(u)\psi(u)$ with any even function $g(u)$ satisfying $g(0) = 1$.

This symmetry relation is closely analogous to the "boundary cross-unitarity relation" in integrable QFT (compare (55) to eq. (3.35) in [30]). We note that relation (55) fixes the first derivative $\psi'(0)$ as

$$\psi'(0) = \check{R}'(0)\psi(0) = \check{R}'(0)(\omega \cdot \omega). \tag{56}$$

The $R$-matrices are polynomials of the rapidity parameter (apart from the normalization factor), therefore it is natural to suspect that the relevant finite dimensional solutions to (49) will be polynomials as well. Then the sq.r.r. has an immediate consequence for the asymptotic behaviour of $\psi(u)$.

**Theorem 6.** *For a given solution let $\alpha$ denote the highest degree of $u$ in $\psi(u)$, and let $\phi_{ab}$ be the coefficient of $u^\alpha$. If the MPS is irreducible then all $\phi_{ab}$ are scalars.*

*Proof.* Let us take the sq.r.r. and take the $u \to \infty$ limit. The asymptotic value of the $\check{R}$ matrices is the permutation operator for both the SU($N$) and the SO($N$) case, therefore we obtain the simple commutativity condition

$$\phi_{ab}\omega_c = \omega_c\phi_{ab}. \tag{57}$$

If there are no irreducible subspaces, then it follows from Lemma (1) that every $\phi_{ab}$ is proportional to the identity matrix in $V_A$. □

It can be seen that the limiting values $\phi_{ab}$ are either symmetric or anti-symmetric in the indices $a, b$. This follows most easily from relation (55). We get a symmetric (or anti-symmetric) $\phi_{ab}$ if its degree $\alpha$ in $\psi_{ab}(u)$ is even (or odd), respectively.

In the simple case of the XXZ spin chain it was already shown in [14, 20] that all one-site product states (corresponding to an MPS with bond dimension 1) are integrable, and they can be obtained from the well known $K$-matrices that solve the usual Boundary Yang-Baxter relation. In Appendix C we show that the $K$-matrices associated to the one-site states indeed satisfy the sq.r.r..

## 3.1 The Boundary Yang-Baxter relation

The twisted Boundary Yang-Baxter relation is

$$K_2(v)R_{21}^{\mathsf{T}}(u+v)K_1(u)R_{12}(v-u) = R_{21}(v-u)K_1(u)R_{12}^{\mathsf{T}}(u+v)K_2(v). \tag{58}$$

Eq. (58) is sometimes also called the BYB relation for soliton non-preserving boundary conditions [61, 62]. The main difference as opposed to the "untwisted" BYB relation is the appearance of the transposition operation. Typically the twisted BYB relation is formulated with different conventions: it might include additional shifts in the rapidities (which can be compensated by a redefinition of $K(u)$), or a different transposition defined as

$$A^t = CA^{\mathsf{T}}C, \tag{59}$$

with $C \in \mathrm{End}(V_j), C^2 = 1$ being some crossing matrix. The action of $C$ can be compensated by a basis transformation, and this is discussed in Section 4. In the present work we use (58) and (61) because these forms are most convenient to treat the MPS.

In those cases when the $R$-matrix itself satisfies a crossing relation of the form

$$R^{\mathsf{T}_1}(u) = C_1 R(-u-\sigma)C_1, \tag{60}$$

with some crossing matrix $C$ and crossing parameter $\sigma \in \mathbb{C}$, the eq. (58) is equivalent to the standard BYB relation. In our examples the $SO(N)$-symmetric $R$-matrix is crossing invariant with $C = 1$ (see (10)), but the $SU(N)$-invariant is not.

Making use of the identification (47) we can write the twisted BYB in the form

$$\check{R}_{12}(v-u)\check{R}_{23}(u+v)(\psi(v) \cdot \psi(u)) = \check{R}_{34}(v-u)\check{R}_{23}(u+v)(\psi(u) \cdot \psi(v)), \tag{61}$$

which is satisfied by the two-site block $\psi(u)$. This is a relation in $V_4 \otimes V_3 \otimes V_2 \otimes V_1 \otimes \mathrm{End}(V_A)$, and it is understood that the $\check{R}$ matrices act on the respective components in the tensor product. For a graphical interpretation of (61) see Fig. 6.

The advantage of the representation (61) over (58) is that certain symmetry properties are more easily identified. In particular, (61) involves the same $R$-matrix. This homogeneity in the exchange relation is the reason why it is always the twisted BYB which is relevant for integrable states of homogeneous spin chains.

In our earlier works [14, 31, 32] the BYB relations were used as a starting point to define integrable initial states. Here we take a different approach, and show that the BYB relation actually follows from the integrability condition under certain conditions. We intend to show that the $K$-matrix, which is obtained as the intertwiner from (43)-(47) indeed satisfies the twisted BYB.

First of all we note that for the special point $u = 0$ the twisted BYB in the form (61) is equivalent to a double application of the sq.r.r.. This follows from the initial condition (50). However, there is a more direct connection valid for arbitrary $u, v$.

Let us consider a double dressing of the MPS $|\Psi_\omega\rangle$ with two transfer matrices. As it was already argued in our previous work [14] it follows from the integrability condition that

$$t(u)t(v)|\Psi_\omega\rangle = \tilde{t}(u)\tilde{t}(v)|\Psi_\omega\rangle. \tag{62}$$

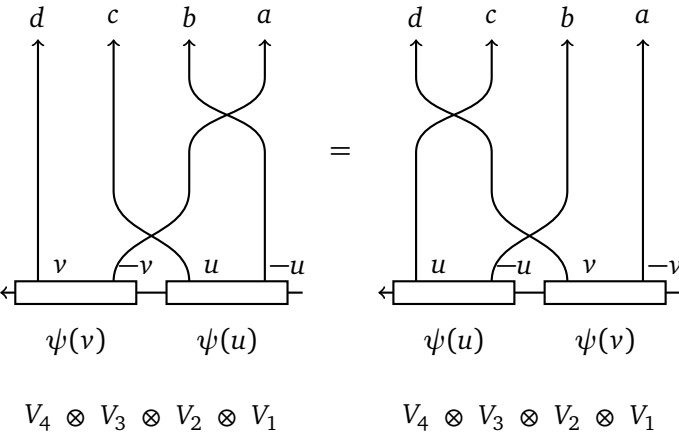

Figure 6: A pictorial representation for the BYB relation for the two-site MPS. $V_{1,2,3,4}$ denote the physical vector spaces, and $a, b, c, d$ are the physical indices. The matrices in the MPS act in the auxiliary space from the right to left. The local $\check{R}$ matrices acting at the crossings are defined such that their argument is always the rapidity coming from the right minus the rapidity coming from the left.

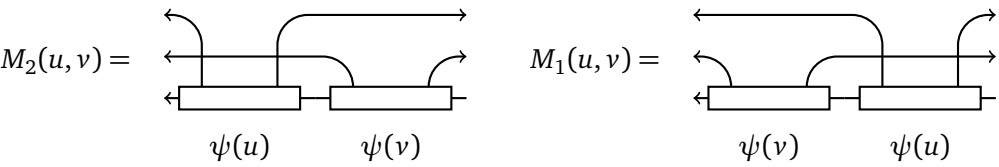

Figure 7: The two intertwiners which are later identified as the two sides of the twisted BYB. Note that the order of $\psi(u)$ and $\psi(v)$ is reversed, but the upper (lower) lines are always associated to the rapidities $u$ and $v$, respectively.

Let us denote the auxiliary product space of the vectors above as $V_{a_1} \otimes V_{a_2} \otimes V_A$, where $V_{a_1}$ and $V_{a_2}$ are the auxiliary spaces for the transfer matrices $t(u)$ and $t(v)$, respectively. We construct the corresponding two sets of dressed matrices as

$$
\begin{aligned}
A_j &= \sum_{k,l} \mathcal{L}_{jk}(u) \otimes \mathcal{L}_{kl}(v) \otimes \omega_l \\
B_j &= \sum_{k,l} \mathcal{L}^{\mathsf{T}}_{jk}(u) \otimes \mathcal{L}^{\mathsf{T}}_{kl}(v) \otimes \omega_l.
\end{aligned}
\tag{63}
$$

Our goal is to construct intertwiners for these doubly dressed MPS:

$$
M(u,v)A_j = B_j M(u,v).
\tag{64}
$$

We consider the following two candidates for the intertwiner $M(u,v)$:

$$
\begin{aligned}
M_1(u,v) &= K_2(v) R^{\mathsf{T}}_{21}(u+v) K_1(u) R_{12}(v-u) \\
M_2(u,v) &= R_{21}(v-u) K_1(u) R^{\mathsf{T}}_{12}(u+v) K_2(v).
\end{aligned}
\tag{65}
$$

Here it is understood that $K_{1,2}(u)$ act on $V_{a_{1,2}}$, respectively. For a graphical interpretation of these two intertwiners see Fig. 7.

Multiple use of the Yang-Baxter relation and the simple intertwiner relation (43) shows that both $M_1(u,v)$ and $M_2(u,v)$ satisfy (64). The idea is that $K_1(u)$ always intertwines the dressing with $\mathcal{L}(u)$, and $K_2(v)$ intertwines $\mathcal{L}(v)$, and the order of the exchange with $K_1(u)$

and $K_2(v)$ is just the opposite for $M_1(u,v)$ and $M_2(u,v)$. We also use that in the intermediate steps the order of the dressings of the MPS can be exchanged as well, using the standard RTT relation. For a graphical interpretation of the steps of the proof see 8.

If the doubly dressed MPS are irreducible then Theorem 1 states that the intertwiners are unique up to normalization, which means that

$$M_1(u,v) = g(u,v)M_2(u,v), \tag{66}$$

with some function $g(u,v)$. Investigating specific components of the relation (61) it can be seen that $g(u,v) \equiv 1$. Thus, given the irreducibility property we have established that the solution of the sq.r.r. also solves the twisted BYB relation (58).

Unfortunately we have not yet managed to prove the twisted BYB independently from the irreducibility property. In our concrete examples we have checked numerically that the doubly dressed MPS are indeed irreducible for generic $u,v$ (see Appendix B), but it would be desirable to establish this by purely analytic means.

Of course it can be checked by direct substitution whether a specific solution to (49) also solves (61). We performed this in our concrete examples (presented later in Sec. 5 and 6) and found agreement.

## 3.2 Implications of integrability: the Quantum Transfer Matrix

Here we show that the BYB relation (61) allows for the construction of commuting sets of double row transfer matrices. The construction is essentially the same as in the papers dealing with soliton non-preserving boundary conditions [62,63]. In the present context these double row objects will be called Quantum Transfer Matrices (QTM's), in analogy with the thermal case [64].

Instead of the homogeneous chain it is useful to introduce an alternating sequence of inhomogeneities $(-u_1, u_1, -u_2, u_2, \ldots, -u_{L/2}, u_{L/2})$. The parameters $u_j$ will play the role of spectral parameters for "double-row transfer matrices" to be constructed.

We define two inhomogeneous transfer matrices as

$$
\begin{aligned}
t(v|u_1, \ldots, u_{L/2}) &= \text{Tr } T(v|u_1, \ldots, u_{L/2}), \\
T(v|u_1, \ldots, u_{L/2}) &= R_{0L}(v - u_{L/2})R_{0,L-1}(v + u_{L/2})\ldots R_{02}(v - u_1)R_{01}(v + u_1) \\
\tilde{t}(v|u_1, \ldots, u_{L/2}) &= \text{Tr } \tilde{T}(v|u_1, \ldots, u_{L/2}), \\
\tilde{T}(v|u_1, \ldots, u_{L/2}) &= R_{01}(v - u_1)R_{02}(v + u_1)\ldots R_{0,L-1}(v - u_{L/2})R_{0L}(v + u_{L/2}).
\end{aligned}
\tag{67}
$$

Here the $L$ inhomogeneities are of alternating signs, and in the notation we write them as $(u_1, \ldots, u_{L/2})$.

We also define inhomogeneous initial states as

$$|\Psi(u_1, u_2, \ldots, u_{L/2})\rangle = \sum_{i_1, \ldots, i_L = 1}^{N} \text{Tr}_0\left[\psi_{i_L, i_{L-1}}(u_{L/2})\ldots \psi_{i_2, i_1}(u_1)\right]|i_L, \ldots, i_1\rangle. \tag{68}$$

The physical states (21) are reproduced in the homogeneous limit $u_j \to 0$ due to the initial condition (50).

Using the unitarity condition we can write (61) in the form

$$\psi(v) \cdot \psi(u) = \check{R}_{23}(-u - v)\check{R}_{12}(u - v)\check{R}_{34}(v - u)\check{R}_{23}(u + v)(\psi(u) \cdot \psi(v)). \tag{69}$$

This exchange relation can be extended to the full MPS, for example

$$
\begin{aligned}
|\Psi(u_1, u_2, \ldots, u_{L/2})\rangle &= \\
\check{R}_{23}(-u_1 - u_2)\check{R}_{12}(u_1 - u_2)&\check{R}_{34}(u_2 - u_1)\check{R}_{23}(u_1 + u_2)|\Psi(u_2, u_1, \ldots, u_{L/2})\rangle,
\end{aligned}
\tag{70}
$$

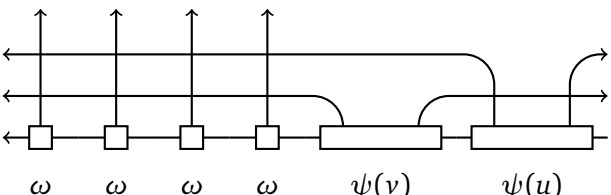

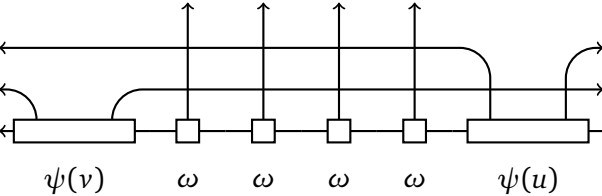

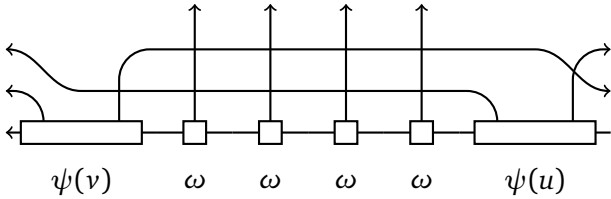

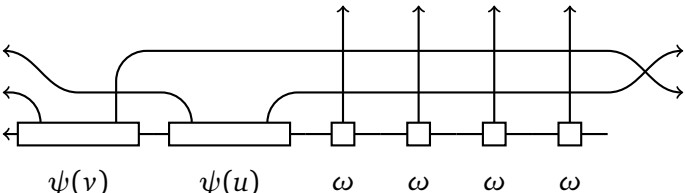

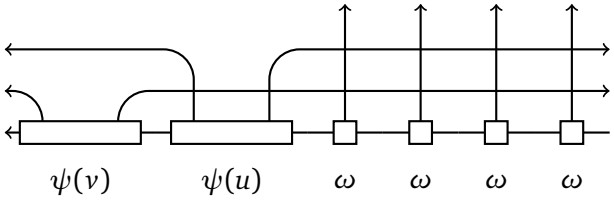

Figure 8: A graphical demonstration for the double intertwining using $M_1(u, v)$, which is the object shown on the r.h.s. of Fig. 7. Here we intertwine multiple products of the doubly dressed MPS, by multiple use of the simple intertwining relation (43) (see Fig. 5) and the Yang-Baxter relations (4). Essentially the same steps (although in a different order) can be repeated also with $M_2(u, v)$, which is shown on the l.h.s. of Fig. 7. If the intertwiner is unique then $M_1$ and $M_2$ have to be proportional to each other. This intertwining relation thus establishes a connection between the twisted BYB and the integrability condition.

and similarly for other exchanges of the inhomogeneity parameters. It follows from the Yang-Baxter equations that this exchange relation is also compatible with the inhomogeneous monodromy matrices, and we have for example

$$
t(v|u_1, u_2, \ldots, u_{L/2})|\Psi(u_1, u_2, \ldots, u_{L/2})\rangle = \check{R}_{23}(-u_1 - u_2)\check{R}_{12}(u_1 - u_2)\times
$$
$$
\times \check{R}_{34}(u_2 - u_1)\check{R}_{23}(u_1 + u_2)\Big[ t(v|u_2, u_1, \ldots, u_{L/2})|\Psi(u_2, u_1, \ldots, u_{L/2})\rangle \Big].
\tag{71}
$$

Similar relations hold for arbitrary products of the transfer matrices[7].

Let us consider partition functions involving two different MPS, that serve as initial and final states. They will be denoted as $|\Psi_A\rangle$ and $\langle\Psi_B|$, and the different subscripts indicate that they are not necessarily adjoints of each other. It is important that both two-site blocks $\psi_A(u)$ and $\psi_B(u)$ satisfy the same twisted reflection relation (61).

We define the inhomogeneous dual MPS vectors as

$$
\big\langle \Psi_B(u_1, u_2, \ldots, u_{L/2}) \big| = \sum_{i_1, \ldots, i_L = 1}^{N} \mathrm{Tr}_0 \Big[ \psi_{B, i_L, i_{L-1}}(-u_{L/2}) \ldots \psi_{B, i_2, i_1}(-u_1) \Big] \langle i_L, \ldots, i_1 |.
\tag{72}
$$

It is important that the rapidity parameters are taken with a sign difference. As an effect these states satisfy the dual exchange relation

$$
\big\langle \Psi_B(u_1, u_2, u_3, \ldots, u_{L/2}) \big| \check{R}_{23}(-u_1 - u_2)\check{R}_{12}(u_1 - u_2)\check{R}_{34}(u_2 - u_1)\check{R}_{23}(u_1 + u_2) =
$$
$$
\big\langle \Psi_B(u_2, u_1, u_3, \ldots, u_{L/2}) \big|,
\tag{73}
$$

and similarly for exchanges of other rapidity pairs.

Let us consider the partition functions

$$
Z_{AB}(v_1, \ldots, v_m | u_1, \ldots, u_{L/2}) =
$$
$$
\big\langle \Psi_B(u_1, \ldots, u_{L/2}) \big| \prod_{j=1}^{m} t(v_j | u_1, \ldots, u_{L/2}) \big| \Psi_A(u_1, \ldots, u_{L/2}) \big\rangle.
\tag{74}
$$

The $Z_{AB}$ are completely symmetric in both the $u$- and the $v$-parameters. Symmetry with respect to $v_j$, $j = 1 \ldots m$ follows from the commutativity of the transfer matrices, whereas symmetry with respect to $u_j$, $j = 1 \ldots L/2$ follows from the above exchange relations involving the initial and final states. For a pictorial interpretation of the partition functions see Fig. 9.

In the physical applications it is usually required that the final state is the dual vector to the initial state, which results in the requirement

$$
\psi_{B, j, k}(0) = (\psi_{A, j, k}(0))^*, \qquad j, k = 1 \ldots N.
\tag{75}
$$

Generally this means that $\psi_A(u)$ and $\psi_B(u)$ are two distinct solutions to (61), unless all the matrices can be chosen as completely real.

The $Z_{AB}$ can be interpreted as the Loschmidt amplitude for some discrete time evolution: in the homogeneous limit $u_j \to 0$ they can be used to approximate the physical Loschmidt amplitude

$$
Z_{AB} \approx \langle \Psi_0 | e^{-iHt} | \Psi_0 \rangle.
\tag{76}
$$

For the details of this procedure we refer the reader to [14].

---

[7]Such exchange relations hold for any transfer matrix which is built from Lax operators satisfying the fundamental exchange relation dictated by the $R$-matrix. They are the so-called "fused transfer matrices", and they also include the space reflected fundamental transfer matrix $\tilde{t}(v|u_1, u_2, \ldots)$, see for example [31, 32]. In this work we don't discuss the fusion hierarchy in detail, therefore the main discussion is limited to the products of the fundamental transfer matrix.

The partition functions (74) allow for an alternative evaluation, which leads to the introduction of the double row Quantum Transfer Matrices, which act in the horizontal direction in Fig. 9. It can be read off Fig. 9 (or it can be established by purely algebraic means) that their explicit form is

$$
\mathcal{T}_{AB}(u|v_1,\ldots,v_m) =
$$
$$
\sum_{a_1,a_2,b_1,b_2=1}^{N} \psi_{A,b_2,b_1}(u) \otimes \left[ T_{a_2 b_2}(-u|-v_1,\ldots,-v_m) T_{a_1 b_1}(u|-v_1,\ldots,-v_m) \right] \otimes \psi_{B,a_2,a_1}(-u).
$$
(77)

Alternatively this can be computed as

$$
\mathcal{T}_{AB}(u) = \mathrm{Tr}\left( M_A(u) K_B^\mathsf{T}(-u) \right),
$$
(78)

where

$$
M_A(u) = T(-u) K_A(u) T^\mathsf{T}(u)
$$
(79)

is the "quantum monodromy matrix", and the products and traces above are to be understood in the indices $a_{1,2}$, $b_{1,2}$ which originally label the states of the physical spin chain.

The partition function is then evaluated as

$$
Z_{AB}(v_1,\ldots,v_m|u_1,\ldots,u_{L/2}) = \mathrm{Tr}\left[ \prod_{j=1}^{L/2} \mathcal{T}_{AB}(u_j|v_1,\ldots,v_m) \right].
$$
(80)

The symmetry properties of $Z_{AB}$ are equivalent to the commutativity condition

$$
[\mathcal{T}_{AB}(u_1), \mathcal{T}_{AB}(u_2)] = 0,
$$
(81)

which can be proven directly using (61) and the Yang-Baxter relations.

We note that depending on the circumstances the double row transfer matrices (77) can be used to define integrable boundary models with local Hamiltonians and additional degrees of freedom at the two boundaries. For an example of such a model see [65]. In Sections 5 and 6 we present explicit examples for the integrable two site block $\psi_{ab}(u)$, which are new in some cases and thus would lead to new integrable boundary conditions. However, it is not the purpose of the present paper to investigate the Hamiltonians and/or the spectra of these models, and this is left to further work.

## 4 Relation with the twisted Yangian

In the following we describe the twisted Yangians that are relevant to the integrable MPS. Our goal is to establish a direct relation to the abstract algebraic setting; in this Section we limit ourselves to the case of the symmetric pair $(\mathrm{SU}(N), \mathrm{SO}(N))$.

The twisted Yangian $Y^+(N)$ corresponding to the orthogonal Lie algebra $\mathfrak{o}(N)$ is an abstract algebra with generators $s_{kl}^{(j)}$, $j = 1, 2, \ldots, \infty$, $k, l = 1, \ldots, N$ given by the following exchange relations [44, 45]. Let us define the matrix $S(u)$ as a formal series in $u^{-1}$ as

$$
S(u) = \sum_{k,l=1}^{N} s_{kl}(u) \otimes E_{kl}, \qquad s_{kl}(u) = \delta_{kl} + \sum_{j} s_{kl}^{(j)} u^{-j}.
$$
(82)

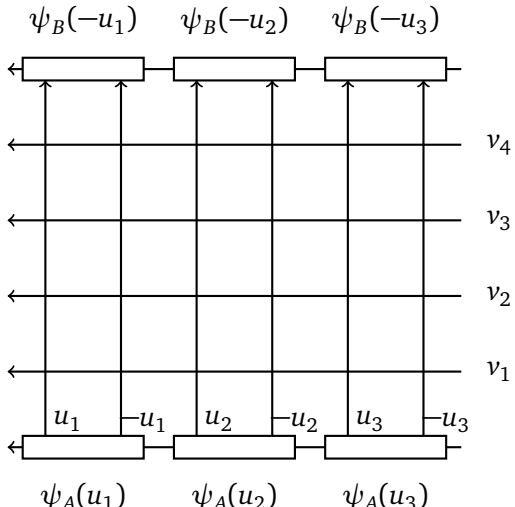

Figure 9: An example for a partition function with integrable boundaries. In the original physical picture the bottom and top rows are interpreted as the MPS which serve as the initial and final states for some discrete time evolution. Alternatively, the partition function can be evaluated by the Quantum Transfer Matrix, which acts in the horizontal direction: in this picture the two-site blocks play the role of integrable boundaries, with an additional degree of freedom at the boundary.

Then the exchange relations are given by the formal equations

$$R(v-u)S_2(v)R^t(-u-v)S_1(u) = S_1(u)R^t(-u-v)S_2(v)R(v-u)$$
$$S^t(-u) = S(u) + \frac{S(u)-S(-u)}{2u}. \tag{83}$$

Here the transposition is defined as in (59) with a charge conjugation matrix given by $C_{j,k} = \delta_{j,N-k+1}$. The second relation is equivalent to our symmetry relation (55) specified to the SU($N$)-symmetric $R$-matrix.

The conventions of the relations (83) are different from the ones used in our (58). The first difference is simply a sign convention for the rapidity parameter, which originates from the convention $R(u) = 1 - u^{-1}P$ used in [46]. This can be compensated immediately. The second difference is that our equations involve only standard transpositions. Nevertheless the two sets of equations can be transformed into each other by a basis transformation: let $Z \in$ SU($N$) be a matrix satisfying

$$ZZ^\mathsf{T} = C. \tag{84}$$

Such a matrix is easily found: in the case of $N = 2$ we can choose

$$Z = \frac{1}{\sqrt{2}}\begin{pmatrix} 1 & i \\ 1 & -i \end{pmatrix}, \tag{85}$$

and for higher $N$ we can build $Z$ analogously using the block structure of $C$. If $S(u)$ satisfies the defining relations (83) then the transformed matrix defined as

$$K(u) = Z^{-1}S(-u)Z \tag{86}$$

satisfies our (58). This can be checked by direct computation, using the group invariance properties (7)-(9) and also $Z^{-1} = Z^\mathsf{T}C$ which follows from $C^2 = 1$.

The definition (82) itself can be interpreted as an asymptotic condition: it is equivalent to the requirement that in $S_{ab}(u)$ (or $\psi_{ab}(u)$) the leading term in $u$ should be $\delta_{ab}$. This requirement is invariant with respect to the basis transformation (86).

The finite dimensional irreducible representations of $Y^+(N)$ have been characterized in [46]. Any such representation gives a concrete solution $\psi_{ab}(u)$ to (61), which will result in an integrable two-site invariant MPS. We will be looking for the subset of solutions which also satisfy the factorizability condition (50). One possible strategy could be to survey all possible irreps and to investigate the possibility of a factorization separately. Instead, we use a different method: we construct concrete solutions to a given $\omega$ by solving the sq.r.r., this is presented in the next two Sections. Nevertheless here we review the main statements of [46], translated into our framework through relations (86) and (47).

First we define Lax-operators that generate the finite dimensional representations of the Yangian. Let $\Lambda$ be a finite dimensional irreducible representation of $\mathfrak{sl}(N)$ acting on the vector space $V_\Lambda$. For the representation of the $E_{ab}$ standard basis matrices we will use the notation $E_{ab}^\Lambda$. We define the Lax operator $\mathcal{L}^{(1,\Lambda)}(u)$ acting on the product of a physical and auxiliary space $V_1 \otimes V_\Lambda$ as

$$\mathcal{L}_{ab}^{(1,\Lambda)}(u) = u\delta_{ab} + E_{ba}^\Lambda, \tag{87}$$

where $a, b = 1 \ldots N$ are interpreted as the physical indices, and the $E_{ab}^\Lambda$ act on the auxiliary space.

From any such Lax operator we can build transfer matrices in the usual way. In the inhomogeneous case discussed above

$$t^{(\Lambda)}(v) = \mathrm{Tr}_\Lambda \Big[ \mathcal{L}^{(L,\Lambda)}(v - u_{L/2}) \mathcal{L}^{(L-1,\Lambda)}(v + u_{L/2}) \ldots \mathcal{L}^{(2,\Lambda)}(v - u_1) \mathcal{L}^{(1,\Lambda)}(v + u_1) \Big]. \tag{88}$$

These transfer matrices commute even for different representations:

$$[t^{(\Lambda)}(v), t^{(\Lambda')}(v')] = 0. \tag{89}$$

Let us denote by $F_{ab} = E_{ab} - E_{ab}^t$ the generators of $\mathfrak{o}(N)$ in the defining representation. Let us take an irreducible representation $\Omega$ of $\mathfrak{o}(N)$ with the matrices $F_{ab}^\Omega$ acting on the vector space $V_\Omega$. Then we can construct the irreducible solution $\psi^\Omega(u)$ to the BYB with auxiliary space $V_\Omega$ as [46]

$$\psi_{ab}^\Omega(u) = \delta_{ab} + F_{ab}^\Omega(u + 1/2)^{-1}. \tag{90}$$

Such solutions can be called "root solutions" as they serve as starting points to construct all finite dimensional representations of the twisted Yangian. They obviously satisfy the group symmetry requirement (53) with $\mathcal{G}' = SO(N)$. The MPS corresponding to a root solution $\psi^\Omega(u)$ will be denoted by $\big| \Psi^\Omega \big\rangle$.

Further non-trivial solutions can be obtained by "dressing" these root solutions with the Lax-operators above. Taking any solution $\psi(u)$ a new solution is constructed as

$$\tilde{\psi}_{ab}(u) = \mathcal{L}_{ac}^{(\Lambda)}(v - u)\mathcal{L}_{bd}^{(\Lambda)}(v + u)\psi_{cd}(u). \tag{91}$$

In terms of the $K$-matrices this dressing is given by

$$\tilde{K}(u) = \mathcal{L}^{(0,\Lambda)}(v - u)K(u)(\mathcal{L}^{(0,\Lambda)}(v + u))^t, \tag{92}$$

where 0 denotes the physical space for the action of the $K$-matrix. In terms of the algebraic setting the dressing (92) describes the co-ideal property of the twisted Yangian within the original Yangian with the usual co-product. For a graphical interpretation of this dressing see the first figure of 10.

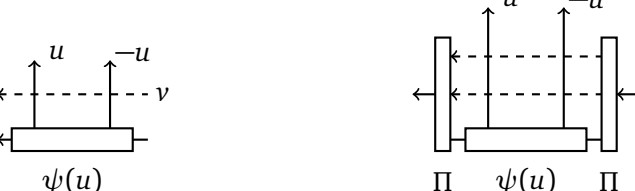

Figure 10: Examples for dressed solutions. The first example involves an "LKL"-type dressing, which on the level of MPS corresponds to the action with a transfer matrix. Generally we can perform multiple instances of these dressings, but the resulting MPS might have invariant subspaces. Irreducible representations of the twisted Yangian are then obtained using a projector $\Pi$ onto an irreducible sup-space in the tensor product of auxiliary spaces.

It is immediately seen that the MPS obtained from the dressed two-site block (91) will be

$$\left|\tilde\Psi\right\rangle = t^{(\Lambda)}(v)\left|\Psi\right\rangle. \tag{93}$$

This idea to construct new integrable MPS by the action of transfer matrices already appeared in [14, 22]. The commutativity of the transfer matrices can be used to obtain an immediate proof of the integrability of the "dressed" MPS.

In principle any finite number of dressings can be performed leading to

$$\tilde K(u) = \left(\prod_{j=1}^{n}\mathcal{L}^{(0,\Lambda_j)}(v_j-u)\right)K(u)\left(\prod_{j=n}^{1}(\mathcal{L}^{(0,\Lambda_j)}(v_j+u))^t\right) \tag{94}$$

and MPS' of the form

$$\prod_{j=1}^{n}t^{(\Lambda_j)}(v_j)\left|\Psi^{\Omega}\right\rangle. \tag{95}$$

However, these MPS are not necessarily irreducible, even if we used irreducible representations $\Omega$ and $\Lambda_j$ of $\mathfrak{o}(N)$ and $\mathfrak{gl}(N)$. A well known example for reducibility is when the product of transfer matrices has invariant subspaces in the tensor product of auxiliary spaces: this happens during the so-called fusion procedure. However, this is not the only possibility: the dressed two-site blocks can have invariant subspaces even if a single product $\prod_{j=1}^{n}\mathcal{L}_{ac}^{(0,\Lambda_j)}(v_j-u)$ constitutes an irreducible representation of the Yangian. The reason is simply that the double product together with the root solution has less symmetries and thus can lead to additional invariant subspaces.

It is proven in [46] that all finite dimensional representations of the twisted Yangian are obtained by the projection method from dressed solutions like (94): to obtain irreducible solutions one needs to project to the subspace generated by the so-called highest weight vector within the tensor product of the various auxiliary spaces. In practical terms this means that each irreducible solution is of the form

$$\tilde K(u) = \Pi\left(\prod_{j=1}^{n}\mathcal{L}^{(0,\Lambda_j)}(v_j-u)\right)K_{\Omega}(u)\left(\prod_{j=n}^{1}(\mathcal{L}^{(0,\Lambda_j)}(v_j+u))^t\right)\Pi, \tag{96}$$

where $\Pi$ is some projector acting on $V_{\Lambda_n}\otimes\cdots\otimes V_{\Lambda_1}\otimes V_{\Omega}$. For a graphical interpretation see the second figure in 10.

An equivalent interpretation of this statement is, that the dressed MPS' can always be expressed as sums of irreducible MPS as

$$\prod_{j=1}^{n} t^{(\Lambda_j)}(v_j) \left| \Psi^{\Omega} \right\rangle = \sum_{j} \left| \Psi_j \right\rangle, \tag{97}$$

and every finite dimensional irreducible MPS satisfying also (55) will be included in one of these sums.

Our goal in this paper is to find those solutions to the BYB that describe our special one-site invariant MPS, listed at the end of Section 2. One possible strategy would be to investigate the general solution (96), to explicitly compute the initial values at $u = 0$, and to identify those solutions that describe the MPS' with a concrete one-site block $\omega$. However, in the present paper we use a different method: we explicitly solve the sq.r.r. starting from a given integrable $\omega$. By the arguments of Section 3 they also satisfy the twisted BYB and the symmetry relations, and they will constitute an irreducible representation of the twisted Yangians.

## 5 Solutions for the symmetric pair $(\mathrm{SU}(N), \mathrm{SO}(N))$

In this section we analyze the solutions of the sq.r.r. (49) in the case of the symmetric pair $(\mathrm{SU}(N), \mathrm{SO}(N))$.

In this case the $R$-matrix is given by (1) and the sq.r.r. can be written in the form

$$\omega_c \psi_{ba} - \psi_{cb} \omega_a = u[\psi_{ca}, \omega_b]. \tag{98}$$

We are looking for polynomial solutions of finite order:

$$\psi_{ab}(u) = \omega_a \omega_b + \sum_{j=1}^{m} \psi_{ab}^{(j)}(u) u^j. \tag{99}$$

The leading coefficient needs to be a scalar with $\mathrm{SO}(N)$ symmetry, so we have

$$\psi_{ab}^{(m)} = \delta_{ab}. \tag{100}$$

In the following we will assume that the $\omega$ matrices are invertible.

Eq. (98) can be used recursively to fix the coefficients. We get

$$\omega_c \psi_{ba}^{(j)} - \psi_{cb}^{(j)} \omega_a = [\psi_{ca}^{(j-1)}, \omega_b], \qquad j = 1 \dots m, \tag{101}$$

where it is understood that $\psi_{ab}^{(0)} = \omega_a \omega_b$ and the highest order equation corresponding to $j = m + 1$ is trivially satisfied with (100).

The linear operator on the l.h.s. of (101) has a null space, and according to Proposition 4 this space is one dimensional with a basis vector given by $\psi_{ab}^{(0)} = \omega_a \omega_b$. It follows that if we find a concrete solution $\psi_{ba}^{(j)}$ the inhomogeneous equation at order $j$ then most the general solution will be

$$\psi_{ab}^{(j)} + \alpha \omega_a \omega_b, \qquad \alpha \in \mathbb{C}. \tag{102}$$

**Proposition 1.** *The linear term is*

$$\psi_{ab}^{(1)} = \omega_b \omega_a + \alpha \omega_a \omega_b, \tag{103}$$

*with some $\alpha \in \mathbb{C}$.*

*Proof.* It is easy to see that the solution with $\alpha = 0$ satisfies (101), thus we get the general solution as given above. $\square$

Simple solutions are obtained by restrictions on the highest degree $m$. In the linear case we get:

**Proposition 2.** *If the $\omega$ matrices are invertible then the only linear solution is (up to overall re-scaling) $\omega_a = \gamma_a$ and*

$$\psi_{ab}(u) = \gamma_a \gamma_b + 2u\delta_{ab}, \tag{104}$$

*where the $\gamma$ matrices satisfy the N dimensional Clifford algebra:*

$$\{\gamma_a, \gamma_b\} = 2\delta_{ab}. \tag{105}$$

*Proof.* In this case the linear term (103) has to coincide with (100), thus

$$\omega_b \omega_a + \alpha \omega_a \omega_b = \beta \delta_{ab}. \tag{106}$$

The r.h.s. is symmetric with respect to $a, b$, yielding $\alpha = 1$. With a proper normalization of $\omega_a$ we can set $\beta = 2$, and thus we obtain the Clifford algebra. $\square$

In this case the solution can be written alternatively as

$$\psi_{ab}(u) = (2u + 1)\tilde{\psi}_{ab}(u), \qquad \tilde{\psi}_{ab}(u) = \delta_{ab} + [\gamma_a, \gamma_b]/4(u + 1/2)^{-1}. \tag{107}$$

Here we recognize the generators of SO($N$) in the spinor representation given by $F_{ab} = [\gamma_a, \gamma_b]/4$, thus this solution is equal to a "root solution" as given by (90). We remind that the corresponding MPS is one of our main examples listed in Section 2.

Now we consider the quadratic case. If $\psi_{ab}(u)$ is of second order, then for $j = 2$ in (101)

$$[\omega_a \omega_c + \alpha \omega_c \omega_a, \omega_b] = \beta [\omega_c \delta_{ba} - \delta_{cb} \omega_a], \tag{108}$$

with some $\alpha, \beta \in \mathbb{C}$, where $\beta \neq 0$ is the coefficient of $\delta_{ab}$. The r.h.s. is anti-symmetric with respect to $a, c$, therefore $\alpha = -1$ or $\{\omega_a, \omega_c\}$ commutes with all $\omega_b$. The latter case leads us back to the Clifford algebra, therefore we are free to set $\alpha = -1$. After an exchange of indices and setting $\beta = -1$ we obtain the exchange relations

$$\left[\omega_a, [\omega_b, \omega_c]\right] = \delta_{ab}\omega_c - \delta_{ac}\omega_b. \tag{109}$$

In this case the second order solution is

$$\psi_{ab}(u) = \omega_a \omega_b - u[\omega_a, \omega_b] - u^2 \delta_{ab}. \tag{110}$$

We construct explicit realizations of the algebra (109). Consider first any triplet of the form $\{\omega_j, \omega_k, \pm i[\omega_j, \omega_k]\}$, $j \neq k$. It follows from (109) that they satisfy the SU(2)-algebra. In those cases when the full set $\{\omega_j\}$ is a Lie-algebra (closed with respect to the commutator), it is easily shown that the SU(2)-algebra is the only possibility, thus we obtain the solution for $N = 3$

$$\psi_{ab}(u) = S_a S_b - u[S_a, S_b] - u^2 \delta_{ab}, \tag{111}$$

where the $S_a$ are the spin operators in a finite dimensional irreducible representation. This solution describes the MPS given by $\omega_a = S_a$, which is one of our main examples listed at the end of Section 2.

In those cases when the set $\{\omega_j\}$ is not a Lie-algebra (the set is not closed with respect to the commutator), we can consider the commutation relations of the matrices $t_{jk} = -t_{kj} \equiv i[\omega_j, \omega_k]$:

$$[t_{jk}, t_{lm}] = -\Big[[\omega_j, \omega_k], [\omega_l, \omega_m]\Big] = t_{jl}\delta_{km} + t_{km}\delta_{jl} - t_{jm}\delta_{kl} - t_{kl}\delta_{jm}. \tag{112}$$

We see that the $t_{jk}$ satisfy the $\mathfrak{o}(N)$ Lie-algebra. Supplied with the additional conditions

$$[\omega_j, t_{kl}] = i[\delta_{jk}\omega_l - \delta_{jl}\omega_k], \qquad [\omega_j, \omega_k] = -it_{jk}, \tag{113}$$

it can be seen that the only irreducible solutions are the "fused spinor representations", which are constructed as follows. Let $\gamma_j$, $j = 1 \ldots N$ be the $N$-dimensional Gamma matrices satisfying the Clifford algebra. Denoting their vector space as $V_\gamma$ we construct the symmetrized tensor product spaces

$$\text{Sym}\left\{\bigotimes_{j=1}^{n} V_\gamma\right\}. \tag{114}$$

Let $\Pi_n$ be the projector from $\bigotimes_{j=1}^{n} V_\gamma$ onto the symmetrized space. Then we construct the matrices

$$\Gamma_j^{(n)} = \Pi_n \left[\underbrace{\gamma_j \otimes 1 \otimes \cdots \otimes 1}_{n} + \underbrace{1 \otimes \gamma_j \otimes \cdots \otimes 1}_{n} + \ldots\right] \Pi_n. \tag{115}$$

Choosing $\omega_j = \Gamma_j^{(n)}/2$ we see that the commutation relations are satisfied, thus we obtain a solution

$$\psi_{ab}(u) = \frac{\Gamma_a^{(n)}\Gamma_b^{(n)}}{4} - u\frac{[\Gamma_a^{(n)}, \Gamma_b^{(n)}]}{4} - u^2\delta_{ab}. \tag{116}$$

Note that this solution has the same form as (111) with the identification $S_a = \Gamma_a^{(n)}/2$. The reason is that the higher spin SU(2) generators are fused symmetrically from the spin-1/2 generators $\sigma_a/2$, and the Pauli matrices satisfy the Clifford algebra.

It can be checked by direct computation that these solutions satisfy the un-normalized symmetry relation (54). For example the solution (104) satisfies (54) with $f(u) = \frac{1+2u}{1-2u}$.

Whereas (104) is a known "root solution" to the twisted BYB, the explicit solutions (111) and (116) are new.

# 6 Solutions for $(\text{SO}(N), \text{SO}(D) \otimes \text{SO}(N-D))$

In this section we analyze the solutions of the sq.r.r. (49) in the case of the symmetric pair $(\text{SO}(N), \text{SO}(D) \otimes \text{SO}(N-D))$, assuming irreducibility.

In this case the $R$-matrix is given by (2), which is crossing invariant by (10). This poses an additional constraint which is most easily derived from the intertwining relation (43). At the special point $u = -c/2$ we have

$$R^{\mathsf{T}}(-c/2) = R(-c/2), \tag{117}$$

therefore the two sets of dressed MPS are completely identical, which by irreducibility implies

$$K(-c/2) \sim I. \tag{118}$$

Furthermore, the crossing and the intertwining relations also imply the inversion relation

$$K(u)K(-u-c) \sim I. \tag{119}$$

If $K(u)$ is a solution to the BYB (58), then the crossed $K$-matrices defined as

$$k(u) = K(-u - c/2) \tag{120}$$

satisfy the standard BYB,

$$k_2(v)R_{21}(u + v)k_1(u)R_{12}(v - u) = R_{21}(v - u)k_1(u)R_{12}(u + v)k_2(v). \tag{121}$$

Condition (118) then translates into the usual initial condition $k(0) \sim 1$.

Instead of a systematic analysis of all possible solutions we construct a few particular ones.

## 6.1 One-site states

Here we assume an MPS with bond dimension one, i.e. a one-site product state given by a vector $\omega_a$.

We make the following Ansatz:

$$\psi_{ab}(u) = f(u)\omega_a\omega_b + g(u)\delta_{ab}, \tag{122}$$

and we assume

$$\omega_a\omega_a = 1. \tag{123}$$

The sq.r.r. reads:

$$(u + c)\omega_c\psi_{ba}(u) + u(u + c)\omega_b\psi_{ca}(u) - u\delta_{cb}\omega_d\psi_{da}(u) = \\ (u + c)\psi_{cb}(u)\omega_a + u(u + c)\psi_{ca}(u)\omega_b - u\delta_{ba}\psi_{cd}(u)\omega_d. \tag{124}$$

This gives the solution

$$f(u) = 2u + c \qquad g(u) = -u. \tag{125}$$

Allowing an arbitrary normalization for $\omega$ we get

$$\psi_{ab}(u) = (2u + c)\omega_a\omega_b - u(\omega_d\omega_d)\delta_{ab}. \tag{126}$$

This shows that in the limit of $\omega_d\omega_d \to 0$ the second term decouples and thus the first one can be chosen as a constant.

After constructing the $K$ matrix as (47) it is easy to see that the inversion relation (119) is satisfied with the following proportionality factor:

$$K(u)K(-u - c) = (-u)(u + c). \tag{127}$$

## 6.2 Two-site states

For sake of completeness we give here the general scalar valued solution of the twisted BYB with the given symmetry properties. This solution describes a two-site state, therefore the sq.r.r. can not be applied here.

The solutions of the BYB relation are classified in [66] and summarized for example in [67]. In our conventions the relevant solution can be written as

$$K(u) = \begin{pmatrix} (N - D - 1 + 2u)I_D & 0 \\ 0 & (-D + 1 - 2u)I_{N-D} \end{pmatrix}, \tag{128}$$

where $I_D$ and $I_{N-D}$ stand for identity matrices of dimension $D$, and $N - D$, respectively. It is easy to see that the inversion relation (119) is satisfied.

In the special case of $D = 1$ we get (a scalar multiple of) the one-site state solution given in (126) with the vector $\omega = (1, 0, \dots, 0)$.

## 6.3 Matrix Product States

To simplify notations we split the full vector space $\mathbb{C}^N$ into a direct sum of $\mathbb{C}^D$ and $\mathbb{C}^{N-D}$. We will use indices $a, b, \ldots = 1 \ldots D$ describing the coordinates in the first component, and $I, J, \ldots = 1 \ldots (N-D)$ for the second.

We investigate the MPS given by

$$\omega_a = \gamma_a \qquad \omega_I = 0, \tag{129}$$

where the $\gamma_a$ are the $D$-dimensional Gamma matrices.

We make the following Ansatz for the solution of the sq.r.r.:

$$\begin{aligned}
\psi_{ab} &= f(u)\gamma_a\gamma_b + g(u)\delta_{ab} \\
\psi_{aI} &= \psi_{Ia} = 0 \\
\psi_{IJ} &= h(u)\delta_{IJ}.
\end{aligned} \tag{130}$$

We now investigate the different components of the sq.r.r. If all indices are in the first subgroup then we get

$$\begin{aligned}
(u+c)\omega_c\psi_{ba}(u) + u(u+c)\omega_b\psi_{ca}(u) - u\delta_{cb}\omega_d\psi_{da}(u) = \\
(u+c)\psi_{cb}(u)\omega_a + u(u+c)\psi_{ca}(u)\omega_b - u\delta_{ba}\psi_{cd}(u)\omega_d.
\end{aligned} \tag{131}$$

Substituting our Ansatz and observing the cancellation of a few terms we get

$$\begin{aligned}
(u+c)\delta_{ab}\gamma_c g(u) + u(u+c)\gamma_b\gamma_c\gamma_a f(u) - u\delta_{cb}\gamma_d(f(u)\gamma_d\gamma_a + g(u)\delta_{ad}) = \\
(u+c)\delta_{bc}\gamma_a g(u) + u(u+c)\gamma_c\gamma_a\gamma_b f(u) - u\delta_{ba}(f(u)\gamma_c\gamma_d + g(u)\delta_{cd})\gamma_d.
\end{aligned} \tag{132}$$

We can use $\gamma_j\gamma_j = D$ to obtain

$$\begin{aligned}
(\delta_{ab}\gamma_c - \delta_{bc}\gamma_a)((2u+c)g(u) + Duf(u)) - \\
+ u(u+C)(\gamma_b\gamma_c\gamma_a - \gamma_c\gamma_a\gamma_b)f(u) = 0.
\end{aligned} \tag{133}$$

Making use of the Clifford algebra relations we end up with

$$(\delta_{ab}\gamma_c - \delta_{bc}\gamma_a)\big((2u+c)g(u) + (-2u^2 - 2cu + Du)f(u)\big) = 0. \tag{134}$$

A solution is

$$f(u) = 2u + c \qquad g(u) = 2u^2 + (-D + 2c)u. \tag{135}$$

For the remainder we only need to check cases when two out of three indices belong to $IJ$, because the other possibilities are automatically zero.

When the indices are specified as $cIJ$ we get

$$(u+C)\gamma_c\delta_{IJ}h(u) = -u\delta_{IJ}(f(u)\gamma_c\gamma_d + g(u)\delta_{cd})\gamma_d = -u\delta_{IJ}(Df(u) + g(u))\gamma_c, \tag{136}$$

giving

$$h(u) = -u(D + 2u). \tag{137}$$

It is easy to see that all other cases are satisfied are well. The solution is thus

$$\begin{aligned}
\psi_{ab}(u) &= (2u+c)\gamma_a\gamma_b + (2u^2 + (-D + 2c)u)\delta_{ab} \\
\psi_{aI}(u) &= \psi_{Ia}(u) = 0 \\
\psi_{IJ}(u) &= -u(D + 2u)\delta_{IJ}.
\end{aligned} \tag{138}$$

It is easy to see that the condition (118) is indeed satisfied.

Setting $D = N$ and substituting $c = N/2 - 1$ we obtain the completely SO($N$)-invariant solution

$$\psi_{ab}(u) = (2u + N/2 - 1)\gamma_a\gamma_b + (2u^2 - 2u)\delta_{ab}. \tag{139}$$

In the special case of $D = 3$ (and arbitrary $N$) the Clifford generators are given by the Pauli matrices. For $N = 6$ we get a particular solution

$$\begin{aligned}
\psi_{ab}(u) &= (2u + 2)\sigma_a\sigma_b + (2u^2 + u)\delta_{ab} \\
\psi_{aI}(u) &= \psi_{Ia}(u) = 0 \\
\psi_{IJ}(u) &= -u(3 + 2u)\delta_{IJ}.
\end{aligned} \tag{140}$$

This solution is relevant to one-point functions in AdS/CFT [24, 26]. After the crossing transformation (120) we get a particular solution $k(u)$ to the untwisted BYB, which was obtained earlier by DeWolfe and Mann in [65].

Direct computation shows that these solutions satisfy the un-normalized symmetry relation (54). For example the simplest solution (126) satisfies (54) with $f(u) = \frac{c+2u}{c-2u}$.

# 7 Conclusions

In this work we treated integrable MPS and established a new linear intertwining relation (43)-(49) called the "square root relation" which guarantees the integrability property. We showed that under certain conditions the solutions to the sq.r.r. also solve the twisted BYB relation (58). The sq.r.r. is only linear as opposed to the quadratic BYB, thus it is much easier to solve. From the general point of view of boundary integrability, we can expect that any solution of the twisted BYB which factorizes at the special point $u = 0$ (see the initial condition (50)) also solves the sq.r.r. with the corresponding one-site object $\omega$.

The question of under which circumstances the three different relations (the integrability condition, and the sq.r.r. and twisted BYB) are completely equivalent remains open. The connection between these three properties is depicted in Fig. 11. We have shown that the three conditions are equivalent if certain dressed MPS are irreducible. However, there are solutions which satisfy all three conditions without the complete reducibility (for example the reference state in the SU($N$)-invariant model), so perhaps this condition can be weakened. It would be desirable to clarify this issue. Also, it would be important to develop analytic proofs for the irreducibility condition, which we confirmed only numerically in a few concrete cases. The reducibility properties of the dressed MPS are related to the reducibility of the fused representations of the twisted Yangian, for which there are no general results available. These particular problems deserve further work.

In Sections 5-6 we have provided solutions of the sq.r.r. for the integrable MPS that were listed at the end of Section 2. Moreover, in 5 we classified all solutions to the sq.r.r. having the symmetry (SU($N$), SO($N$)) which are at most quadratic polynomials in the rapidity. The linear solutions necessarily lead the Clifford algebra, whereas the quadratic solutions correspond to symmetrically fused Clifford generators. On the other hand, this list does not exhaust all known integrable MPS: the fused states for the pair (SO($N$), SO($D$) ⊗ SO($N-D$)) (see [24–26] for details) were not treated here.

Having found our solutions to the sq.r.r., and having established that they also solve the twisted BYB there are the following open directions.

First, one should establish the explicit fusion hierarchy of these solutions, which could lead to an understanding of the physical overlaps using the combination of the Quench Action and QTM methods. These computations would be analogous to the ones of [20, 32].

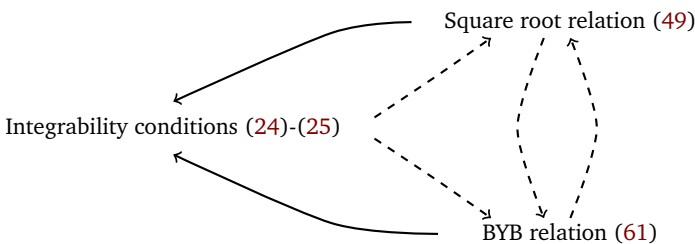

Figure 11: Relation between the different notions of integrability for a one-site invariant MPS. The thick lines stand for direct implication, whereas the dashed lines show implication given the irreducibility condition of the dressed MPS, as explained in the main text.

Second, one should understand better the explicit relations with the representation theory of the twisted Yangian. In particular, it would be useful to understand the exact decomposition of the form (96), together with the projector $\Pi$. Preliminary computations show that our solution (111) can always be obtained by a single fusion of LKL type and a non-trivial projection. Determining the decomposition could also help in understanding the overlap formulas obtained in [23, 24, 26].

Finally, it would be interesting to obtain the spectrum and the eigenstates of the double row transfer matrices with the new boundary $K$-matrices. As we remarked, these transfer matrices can lead to new integrable models with local Hamiltonians and additional boundary degrees of freedom.

We plan to return to these questions in further research.

## Acknowledgments

We would like to thank Tamás Gombor, Ben Grossmann, Charlotte Kristjansen, Marius de Leeuw, Georgios Linardopoulos, Vidas Regelskis, Gábor Takács, Kasper Vardinghus and Matthias Wilhelm for useful discussions. BP was supported by the BME-Nanotechnology FIKP grant of EMMI (BME FIKP-NAT), by the National Research Development and Innovation Office (NKFIH) (K-2016 grant no. 119204 and KH-17 grant no. 125567), and by the Premium Postdoctoral Program of the Hungarian Academy of Sciences. EV acknowledges support by the EPSRC under grant EP/N01930X.

## A The different forms of the square root relation

Here we provide a detailed derivation showing that (48) follows from (43).
We start with the intertwining relation

$$A_j K(u) = K(u) B_j, \quad j = 1, \ldots, N, \tag{141}$$

which is a relation in $\text{End}(V_0 \otimes V_A)$ and it concerns the dressed MPS defined as

$$
\begin{aligned}
A_j &= \sum_k \mathcal{L}_{jk}(u) \otimes \omega_k \\
B_j &= \sum_k \mathcal{L}_{jk}^{\mathsf{T}}(u) \otimes \omega_k.
\end{aligned}
\tag{142}
$$

Here the Lax operators $\mathcal{L}_{jk}(u)$ are defined from the expansion of the $R$-matrix as

$$R_{10}(u) = \sum_{ab} E_{ab} \otimes \mathcal{L}_{ab}(u), \tag{143}$$

where $E_{ab}$ are the basis matrices acting on a physical space $V_1 \approx \mathbb{C}^N$ and the matrices $\mathcal{L}_{ab}(u)$ act on the auxiliary space $V_0$. This expansion also implies that the matrix elements of the $R$-matrix are

$$R^{ad}_{bc}(u) = (\mathcal{L}_{ab}(u))^d_c = \mathrm{Tr}_0\big(E_{dc}\mathcal{L}_{ab}\big). \tag{144}$$

In our cases the $R$-matrix is symmetric with respect to an exchange of the two vector spaces, and also with respect to the exchange of the in- and out-states, therefore

$$R^{ad}_{bc}(u) = R^{da}_{cb}(u) = R^{bc}_{ad}(u). \tag{145}$$

We also expand the $K$-matrix as

$$K(u) = \sum_{a,b} E_{ab} \otimes \psi_{ab}(u), \tag{146}$$

where $E_{ab}$ are elementary matrices acting on $V_0$ and $\psi_{ab}(u)$ are matrices acting on $V_A$.

Then for every $j = 1\ldots N$ we get

$$\sum_{a,b,k} \big(\mathcal{L}_{jk}(u)E_{ab}\big) \otimes \big(\omega_k \psi_{ab}(u)\big) = \sum_{a,b,k} \big(E_{ab}\mathcal{L}^\mathsf{T}_{jk}(u)\big) \otimes \big(\psi_{ab}(u)\omega_k\big). \tag{147}$$

We multiply this equation with $E_{cd} \otimes 1$, where again $E_{cd}$ acts on $V_0$ and $c,d = 1\ldots N$ are unspecified indices. Fruthermore, we take the partial trace over $V_0$ to arrive at

$$\sum_{a,b,k} \mathrm{Tr}\big(E_{cd}\mathcal{L}_{jk}(u)E_{ab}\big)\big(\omega_k \psi_{ab}(u)\big) = \sum_{a,b,k} \mathrm{Tr}\big(E_{cd}E_{ab}\mathcal{L}^\mathsf{T}_{jk}(u)\big)\big(\psi_{ab}(u)\omega_k\big). \tag{148}$$

This can be written as

$$\sum_{a,k} \mathrm{Tr}\big(\mathcal{L}_{jk}(u)E_{ad}\big)\big(\omega_k \psi_{ac}(u)\big) = \sum_{b,k} \mathrm{Tr}\big(\mathcal{L}_{jk}(u)E_{bc}\big)\big(\psi_{db}(u)\omega_k\big), \tag{149}$$

where we also used $E^\mathsf{T}_{ab} = E_{ba}$ for every $a,b = 1\ldots N$.

Using (144) and the symmetries of the $R$-matrix this gives

$$\sum_{a,k} R^{ka}_{jd}(u)\big(\omega_k \psi_{ac}(u)\big) = \sum_{b,k} R^{kb}_{jc}(u)\big(\psi_{db}(u)\omega_k\big). \tag{150}$$

Introducing also the matrix $\check{R}(u) = PR(u)$ this is written as

$$\sum_{a,k} \check{R}^{ka}_{dj}(u)\big(\omega_k \psi_{ac}(u)\big) = \sum_{b,k} \check{R}^{bk}_{jc}(u)\big(\psi_{db}(u)\omega_k\big). \tag{151}$$

After an exchange of indices this is indeed equivalent to (48).

# B  Numerical tests of the irreducibility condition

Here we describe our simple numerical procedure to test the irreducibility condition for the dressed MPS defined in (41).

Given a set of $d \times d$ matrices $\{A_j\}_{j=1...N}$ the goal is to test whether the linear span of all matrix products of length $L$ becomes $\text{End}(\mathbb{C}^d)$ for $L \geq L^*$. This can be tested recursively in $L$. At $L = 1$ we investigate the matrices themselves and construct an orthonormal basis using the scalar product

$$(A, B) \equiv \text{Tr}\, A^\dagger B. \tag{152}$$

Let us denote this basis by $\{B_j^{(1)}\}_{j=1...n_1}$, where $1 \leq n_1 \leq N$. Now we construct the basis at length $L = 2$ by computing all products

$$A_j B_k^{(1)}, \quad j = 1 \ldots N, \quad k = 1 \ldots n_1 \tag{153}$$

and performing a new orthogonalization procedure. This gives a basis $\{B_j^{(2)}\}_{j=1...n_2}$, where typically $n_2 > n_1$. We continue this procedure such that at each step we define

$$\{B_j^{(L+1)}\}_{j=1...n_{L+1}} \equiv \text{basis of span}\{A_j B_k^{(L)}\}_{j=1...N, k=1...n_L}. \tag{154}$$

Generally we observe that $n_L$ is always increasing, and the MPS is irreducible if $n_L$ it reaches the maximal value $d^2$ at some $L = L^*$.

We performed this test for a few examples of the dressed matrices (41) constructed from the integrable MPS listed at the end of Section 2; we took at least one example from each family. We observed that the dressed MPS are irreducible for generic values of the rapidity $u$. On the other hand, there are some special rapidity points (for example $u = 0$), when $n_L$ saturates below $d^2$ and thus the MPS has invariant subspaces. This does not affect the applicability of Theorem 1, because the existence of the unique integrable $K$-matrices is established for almost all $u$, and the intertwiner relation (43) holds even at the special points by continuity.

We observed that the set of special rapidity points for which $n_L < d^2$ always consists of integers. The point $u = 0$ is always included in this set, which can be tied to the special value of the $R$-matrix $R(0) = P$ used in the dressing (41). However, depending on $\{\omega_j\}$ there can be other reducible points. For example in the case of $(\text{SU}(3), \text{SO}(3))$ and $\omega_j = S_j$ with the $S_j$ being the spin-1 generators we find that the reducible points are $u = \{-2, 0, 1\}$.

We also investigated the doubly dressed matrices (63). We observed, that for generic rapidity parameters $u, v$ the MPS are irreducible in all our examples. This implies that the solutions of the sq.r.r. also solve the twisted BYB relation. Once again we observed special points where the MPS have invariant subspaces, but this does not alter the conclusions.

# C  The square root relation in the XXZ chain

Here we prove that the known solutions of the BYB in the XXZ chain also satisfy the sq.r.r., given that we choose those solutions which describe one-site states after the crossing transformation. This way we can also prove that all one-site states are integrable in the XXZ chain, even in odd volumes [20].

In the case of the XXZ chain the crossing transformation for the boundary two-site block is [14]

$$\psi_{ab}(u) = (K(-\eta/2 - u)\sigma_y)_{ab}. \tag{155}$$

The $K$-matrices are given by a known 3-parameter family [68]. Choosing a particular parametrization with constants $\alpha, \beta, \theta$ they lead to the following form of the two-site block $\psi_{ab}(u)$:

$$
\begin{aligned}
\psi_{11}(u) =& -e^{\theta} \sinh(\eta + 2u) \\
\psi_{12}(u) =& 2(-\sinh(\alpha)\cosh(\beta)\cosh(\eta/2 + u) + \cosh(\alpha)\sinh(\beta)\sinh(\eta/2 + u)) \\
\psi_{21}(u) =& 2(\sinh(\alpha)\cosh(\beta)\cosh(\eta/2 + u) + \cosh(\alpha)\sinh(\beta)\sinh(\eta/2 + u)) \\
\psi_{22}(u) =& e^{-\theta} \sinh(\eta + 2u).
\end{aligned}
\tag{156}
$$

One-site states are obtained by choosing $\alpha = 0$ and $\beta = i\pi/2 + \eta/2$. Dividing by $\sinh(\eta)$ we obtain the two-site block

$$
\begin{aligned}
\psi_{11}(u) =& -e^{\theta} \frac{\sinh(\eta + 2u)}{\sinh(\eta)} \\
\psi_{12}(u) =& \psi_{21}(u) = i \frac{\sinh(\eta/2 + u)}{\sinh(\eta/2)} \\
\psi_{22}(u) =& e^{-\theta} \frac{\sinh(\eta + 2u)}{\sinh(\eta)},
\end{aligned}
\tag{157}
$$

satisfying the initial condition

$$
\psi(0) = \omega \cdot \omega, \quad \text{with} \quad \omega = \begin{pmatrix} ie^{\theta/2} \\ e^{-\theta/2} \end{pmatrix}.
\tag{158}
$$

It can be checked by direct computation that this $\psi(u)$ solves the sq.r.r. with the $\omega$ given above.

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
