# Peer review of "Integrable Matrix Product States from boundary integrability"

_SciPost Physics, doi:SciPost Phys. 6, 062 (2019)_

## Round 2 · Referee Report · Anonymous (Referee 1) · 2019-3-20

Strengths

1-The paper fills in a gap in the understanding of integrable matrix product states by linking them to twisted Yangians.
2-The authors identify a new reflection-type equation called the square root relation which is related to integrability of the matrix product states and leads to the usual boundary YBE.
3-The result lead to new representation of some twisted Yangians
4-The description in terms of symmetric pairs gives a nice unifying framework for many known integrable states

Weaknesses

1-There are several classifications of integrable Matrix product states now. A clearer picture should be formulated.
2-Not all integrable matrix product states fall into the class that the authors discuss.
3-Unclear if this formulation will help finding overlap functions between MPS and Bethe states

Report

The paper is well written and interesting. It introduces an interesting new relation called the square root relation which is a linear relation which, under certain conditions, leads to solutions of the boundary Yang-Baxter equation. It leads to new representations for twisted Yangians and gives an algebraic framework for a large class of integrable Matrix product states that arise in condensed matter and AdS/CFT. The authors prove some general theorems for states which satisfy the square root relation and work out several examples. The results in the paper are new and of great interest to the community. I recommend this paper for publication, but would like the authors to clarify a few points.

Requested changes

1-At this point there are 3 characterizations of sort of integrable MPS. There is (2.24), there is the relation to the bYBE and there are the MPS that satisfy the reflection equation. I would like the authors to add maybe a diagram with the 3 notions and make clear what their interrelations are. It is described in the text, but a diagram would add much clarity. 2-The authors consider MPS that are invariant under some group action. Indeed, if (2.23) is satisfied, then this is the case. Can the authors comment on the other implication. Are all invariant states of the form (2.23)? 3-Below (3.3), the authors mention that they restrict to matrices of block-diagonal form. In general there can be an upper triangular part F, which does not affect the MPS. This is for example the case in MPS that appear in AdS/dCFT for higher representations [22], which might be cited as an example. Moreover, this implies that for a given MPS, omega is only defined up to an equivalence relation. Of course, the choice F=0 can be made and will simplify the computations. However, can the authors comment if maybe a non-trivial choice of F is needed to show that omega solves the bYBE or the square root relation? 4-Above (3.8) I would like the authors to clarify that the matrices A,B correspond to the MPS and the reflected MPS. It is at that point not really clear what corresponds to what. 5- Above Theorem 6, the authors mention that R-matrices are polynomials of the rapidity parameter, but this is only the case for rational R-matrices. Shortly after they discuss the XYZ spin chain, where the R-matrix is an infinite series in u. Can theorem 6 be extended to this case?

---

## Round 2 · Referee Report · Anonymous (Referee 2) · 2019-3-25

Strengths

1-A theory of matrix product states is linked to the boundary YBE and to representation theory of twisted Yangians.
2-An interesting reformulation of the boundary intertwining equation and the boundary YBE is proposed.
3-A number of new solutions to a three-legged twisted boundary YBE is found.

Weaknesses

2-The notation used in the paper is not well explained.
3-Some steps in deriving key identities are omitted.

Report

In this paper the authors are studying the integrable Matrix Product States (MPS) in integrable spin chains and establish a new linear intertwining relation, which they call the 'square root relation' (sq.r.r.). The authors demonstrate that under certain conditions the solutions to the sq.r.r. are also solutions to the twisted reflection equation. The authors then provide explicit examples of such solutions which were not explicitly know before. These solutions may be viewed as concrete realizations of finite-dimensional representations of certain twisted Yangians and are interesting on their own right. Overall, this is an interesting paper and deserves to be published after the authors have addressed the technical points raised below.

Requested changes

1-Does the star $*$ mean complex conjugation in (2.12)?

2-Below (2.20) $d$-dimensional matrices $\omega^{(j)}$ are introduced. In (2.21) the notation is changed to $\omega_{j_k}$. Are these the same objects? Since $\omega^{(j)}$ act on $V_A$, is $V_A$ a $d$-dimensional vector space?

3-Below (2.22) a representation $\Lambda_\omega$ of $\mathcal{G}'$ in the space $V_A$ is introduced. However such a representation does not necessary need to exist unless some assumptions on $V_A$ are introduced a priori.

4-The object $G_{jk}$ in (2.23) is not defined. It is then used many times throughout the paper but I was unable to find its definition and thus verify those relations.

5-Page 6, line 6: please clarify the representation of $\mathcal{G}'$ assumed in the tensor product $\Lambda_\omega \otimes \bar\Lambda_\omega \otimes \mathcal{G}'$. I expect that this should be $\Lambda_0$ introduced below equation (3.9). It might be helpful to introduce this notation earlier.

6-Page 7, line 3: in *a* finite dimensional.

7-Page 8, line 2: an invariant su*b*-space.

8-Below (3.15) an object $\psi(u)$ is introduced without a definition. Please be more specific what it is and what are the spaces it acts on. Likewise, in (3.16) the notation $\omega$ is used. It was stated above (2.22) that $\omega$ will denote a collection $\{\omega_j\}$. This does not seem to be right. Also, below (3.16) it is stated that this relation acts on $V_3\otimes V_2 \otimes V_1$, however $\psi(u)$ and $\omega$ are objected acting on $V_A$. I must be misunderstanding something. Please clarify equation (3.16). Lastly, it is not that obvious that (3.11) is equivalent to (3.16). It would help the reader if you could provide more details since this is a key relation in this paper.

9-Below (3.17): perhaps it would better to say 'up to a scalar factor'. A 'trivial numerical factor' suggests that it is $0$ or $1$.

10-Page 12, line 2 of the proof of Theorem 4: it would help the reader if you could provide details of how to obtain this relation from (3.16). Such manipulations are used throughout the paper and it would be helpful to demonstrate them at least once.

11-Below (3.20): It is stated that $\psi(0) = \omega \otimes \omega$. However, I had an impression from above that $\psi(u) \in {\rm End}(V_A)$. Please clarify this point.

12-Equations (3.21), (3.22) and many more below can not be verified without $\psi(u)$ being properly defined. Please see my comments 8 and 11 above.

13-Equation (3.28) is stated to be equivalent to (3.25). It would help the reader if you could provide details of how to obtain (3.28) from (3.25).

14-Page 28, line 10: Here we describe *a* simple.

15-Page 28, line 13: Please explain object $L^*$.

---

## Round 3 · Referee Report · Anonymous (Referee 1) · 2019-5-2

Report

The authors have addressed the comments in a satisfactory way.

---

## Round 3 · Referee Report · Anonymous (Referee 2) · 2019-5-12

Report

The authors have addressed all the issues I have raised and added an additional Appendix explaining the derivation of (3.17) which they have now explicitly stated in (3.16). However I still find it difficult to understand equation (3.17). First, I believe there should be no tensor product in the r.h.s. of (3.16). This follows from (A.11): matrices $\psi_{ab}(u)$ and $\omega_k$ act on the same auxiliary space $V_A$. Second, tensor products $\omega \otimes \psi(u)$ and $\psi(u) \otimes \omega$ of collections of matrices $\omega_k$ and $\psi_{ab}(u)$ in (3.17) act on a tensor product $V_A \otimes V_A$ contradicting to what is stated right below, and so (3.17) is not equivalent to (3.16). Please correct this issue.

Requested changes

1) p.8, line 2, and p.28, line 5 from the bottom: $V_A\otimes V_0$ should be $V_0\otimes V_A$. 2) a tensor product symbol is missing in (3.8) and (A.3). 3) p.28, line 5 from the bottom: concernes should be concerns.

---

## Round 3 · Author Response

We are thankful to the referees for the comments. We believe that we made all the requested changes, and also, we think that this made the manuscript better. For a list of changes see below.

---

## Round 3 · List of Changes

Referee 1.

  1. We included a diagram at the Conclusions, with proper referencing. We hope that the relations are clear now.
  2. We are especially thankful for this question, because we did not consider it before, and it is actually nice. Now we added a footnote about this: The later theorem in the text can be used to show that indeed all such states are of this form (they enjoy local group invariance), if the MPS is group invariant in any volume, and it is completely reducible. We did not include a full detailed proof of this claim. On the one hand side it is not really important, on the other hand the proof would use material which is presented only later in the text. So we just added this footnote.
  3. There can be cases with non-trivial F parts, but we did not treat such cases here. We added some comments about this.
  4. We clarified this.
  5. At the moment we have not yet investigated the XYZ case, so we don't know the answer to this question. The present paper only deals with the polynomial cases. Perhaps later research can clarify this.

Referee 2

  1. Yes and we explained it.
  2. Indeed there was a mismatch in the notations, we tried to clarify it.
  3. Indeed, and we modified that part of the sentence.
  4. Here G stands for the matrix in the defining representation, we added the explanation in the text.
  5. We explained this now, the G' representation is inherited from the defining rep. of the full original group.
  6. Corrected.
  7. Corrected.
  8. This is a point, where we hoped that our manuscript was clear, but the comments of the referee show that it was not clear. So we have now added a few more explanations. The $\psi(u)$ is a four-legged object, so it has 4 indices. This is not written out explicitly, we just state that $\psi_{ab}(u)$ are matrices acting on $V_A$. And we regard $\psi(u)$ as an element of $V_1\otimes V_2\otimes End(V_A)$. We explained in the text now, we hope it is clear. Also, in the Appendix we added a detailed derivation with the indices spelled out, that the two forms of the sq.r.r. are the same.
  9. Corrected.
  10. See point 8. In the Appendix there is one example for this.
  11. We replaced the notation here.
  12. See above.
  13. See point 8 and the new Appendix.
  14. Corrected.
  15. Here $L^*$ is just some threshold value.

---

## Round 4 · Author Response

We have made the requested modifications. Instead of the tensor product notation at that specific point asked by the referee 1 we introduced the $\cdot$ notation. It is important that this is just a short-hand notation, and this is explained in the text. The ,,square root relation'' and the ,,Boundary Yang-Baxter equation'' for the objects $\omega$ and $\psi(u)$ are to be understood with all their indices spelled out, but we believe that sometimes it is useful to write down simplified notations too. We hope that the present form is clear.

---

## Round 4 · List of Changes

-We corrected the typos.
-We replaced the tensor product notation with $\cdot$ at the places requested.

---

## Editorial Decision

published